# Unsupervised Multi-Scale Gromov-Wasserstein Hypergraph Alignment

## Abstract

We consider the problem of unsupervised hypergraph alignment, where the goal is to infer node correspondence between two hypergraphs based solely on their structure. Hypergraphs generalize graphs by allowing hyperedges to connect multiple nodes, and they provide a natural framework for modeling complex higher-order relationships. We introduce FALCON, a framework that effectively unifies hypergraph filtration with a multi-scale Gromov-Wasserstein consensus to solve unsupervised hypergraph alignment. The multi-scale, hierarchical structure revealed by filtration provides a set of robust, nested geometric constraints that are naturally regularized and aggregated by the GW framework. This synergy is uniquely suited to overcoming structural noise, a critical challenge where prior methods fail. Experiments on real-world datasets demonstrate that FALCON substantially outperforms state-of-the-art baselines, proving especially robust to noise.

## 1 Introduction

Graph alignment seeks to identify a correspondence between the nodes of two graphs so that structural properties are preserved. The problem is **NP**-hard and closely related to the Quadratic Assignment Problem (QAP) (Lawler, 1963), making the development of scalable and accurate algorithms particularly challenging (Conte et al., 2004; Foggia et al., 2014; Yan et al., 2016; Tang et al., 2025; Trung et al., 2020). Nonetheless, graph alignment remains a core task in data mining, with wide-ranging applications in image processing, pattern recognition, social network analysis, and bioinformatics (Bunke, 2000; Sun et al., 2020; Haller et al., 2022; Conte et al., 2004; Foggia et al., 2014; Yan et al., 2016). While most research has focused on conventional graphs, real-world systems, such as biological interaction networks or multi-user communication platforms, often involve higher-order interactions. These interactions are naturally captured by hypergraphs (Kim et al., 2024; Lee et al., 2025). Aligning such structures poses additional challenges due to the combinatorial complexity of higher-order interactions. Moreover, in many settings, node features are unavailable or unreliable, necessitating fully unsupervised methods that infer alignment purely from the network topologies.

In this work, we address the problem of *unsupervised hypergraph alignment*, where we seek to recover a meaningful correspondence between the nodes of two hypergraphs without relying on labeled training data, or node/hyperedge features. In this setting, given two hypergraphs, the objective is to maximize the number of correctly aligned nodes with respect to an unknown ground truth.

To solve this problem, we introduce FALCON (**F**iltration-based hypergr**A**ph a**L**ignment via **C**onsensus **O**ptimal tra**N**sport), a fully unsupervised alignment algorithm that operates directly on the hypergraphs and leverages their structural information across multiple scales. Our approach builds on the Gromov-Wasserstein (GW) discrepancy, originally defined for comparing metric measure spaces via optimal transport (Peyré et al., 2016). We formulate the Multi-Scale Gromov-Wasserstein (MSGW) consensus for hypergraph alignment and show it is equivalent to computing a Euclidean Fréchet mean of transport plans, lending it theoretical stability and optimality guarantees.

To further incorporate the multi-scale perspective into the hypergraph alignment problem, we adapt the concept of filtration, a commonly used tool in persistent homology (Aktas et al., 2019; Pun et al., 2022). This approach enables us to construct subhypergraphs at different scales, facilitating a systematic comparison of hypergraphs across multiple levels of abstraction.

FALCON combines MSGW and hypergraph filtration to obtain transport plans at each scale, which are jointly aggregated into a global alignment via a consensus transport matrix. Our multi-scale problem formulation enhances robustness to structural noise and preserves global consistency across filtration levels. In contrast to hypergraph alignment based on clique or bipartite expansions (which transform the hypergraphs to conventional graphs and apply graph-based alignment), our framework directly aligns native hypergraph structures without sacrificing higher-order information. Experiments on real-world datasets demonstrate that FALCON substantially outperforms state-of-the-art baselines, proving especially robust to structural perturbations.

## 2 Related Work

There are several surveys on graph alignment (Conte et al., 2004; Foggia et al., 2014; Yan et al., 2016; Tang et al., 2025; Trung et al., 2020), and a wide range of methods exploit the close connections to graph isomorphism and the quadratic assignment problem (QAP) (Lawler, 1963; Yan et al., 2020). Applications span computational biology (Ma & Liao, 2020), image processing (Sun et al., 2020), social-network de-anonymization and linkage (Senette et al., 2024; Shu et al., 2017), and natural language processing (Osman & Barukub, 2020). A broad spectrum of techniques has been explored, including spectral methods (Hermanns et al., 2023; Nassar et al., 2018), random walks (Cho et al., 2010), probabilistic models (Qi et al., 2021), and others.

Graph alignment methods are typically classified into restricted and unrestricted approaches (Skitsas et al., 2023). Restricted methods rely on partial ground-truth correspondences or additional domain-specific features. For instance, social network linkage methods often incorporate user attributes and partially-known mappings (Nie et al., 2016; Li et al., 2024; Senette et al., 2024), and protein-protein interaction network aligners often rely on biological features (Devkota et al., 2024; Kalaev et al., 2008; Liao et al., 2009; Singh et al., 2008).

In contrast, unrestricted methods operate in an unsupervised setting, using only network topology. Our approach belongs to this category. REGAL (Heimann et al., 2018) aligns graphs via representation learning and embedding alignment. Xu et al. (2019b) propose a framework based on the Gromov-Wasserstein discrepancy to jointly learn node embeddings and transport maps. SGWL (Xu et al., 2019a) improves scalability by recursively partitioning graphs before alignment. CONE (Chen et al., 2020) optimizes neighborhood consistency, computed via Jaccard similarity, and aligns node embeddings accordingly. GRASP (Hermanns et al., 2023) draws on functional maps and heat kernels from shape analysis. PARROT (Zeng et al., 2023) combines optimal transport with restart-based random-walk costs to incorporate both structure and attributes. FUGAL (Bommakanti et al., 2024) proposes an unrestricted graph alignment framework that directly optimizes a relaxed QAP while incorporating a feature-based linear assignment problem (LAP) regularizer. BIGALIGN (Koutra et al., 2013) focuses on the alignment of bipartite graphs by proposing an iterative optimization framework that finds soft correspondence matrices for both node partitions simultaneously.

While most existing alignment methods are limited to pairwise graphs, hypergraphs offer a richer framework for modeling higher-order interactions. Several recent surveys discuss learning on hypergraphs (Gao et al., 2020; Çatalyürek et al., 2023; Antelmi et al., 2023), but alignment techniques remain limited. Tan et al. (2014) study restricted user alignment in hypergraphs using partial correspondences. Mohammadi et al. (2016) extend alignment to graph triangles, generalizing from nodes to higher-order substructures. Do & Shin (2024) present an unsupervised approach using Struct2Vec, contrastive learning, and a graph adversarial network to match hypergraph embeddings.

Further work on higher-order alignment appears in computer vision, where methods often assume $k$-uniform hyperedges, rely on geometric features, and use partial correspondences (Nguyen et al., 2015). For example, CURSOR is a feature-driven approach designed for low-order, uniform or near-uniform hypergraphs (Zheng et al., 2024). Other supervised approaches frame hypergraph matching as a node classification task (Liao et al., 2021). Additional works repurpose hypergraph structures for related tasks, e.g., $H^2MN$ (Zhang et al., 2021) for graph similarity and attention-based scoring, or target tracking with rule-based label disambiguation (Li et al., 2023).

In contrast, our FALCON algorithm is fully unsupervised and infers node correspondences by minimizing a multi-scale Gromov-Wasserstein discrepancy over structural topology alone.

## 3 PRELIMINARIES

Table 7 in the appendix gives an overview of the used notation. We use $[k]$ with $k \in \mathbb{N}$ to denote the set $\{1, \ldots, k\}$ and we write $\Delta_k := \left\{ w \in \mathbb{R}^k_{\geq 0} \mid \sum_{m=1}^k w_m = 1 \right\}$ for the probability simplex of dimension $k$. An undirected *hypergraph* $G = (V, E)$ consists of a finite set of nodes $V$ and a finite set of *hyperedges* $E \subseteq 2^V \setminus \{\emptyset\}$, i.e., each hyperedge $e \in E$ is a non-empty subset of $V$. Given a hypergraph $G = (V, E)$ and a subset of hyperedges $E' \subseteq E$, the subhypergraph induced by $E'$ is defined as $G' = (V', E')$ where $V' = \bigcup_{e \in E'} e$. We define the cardinality of an edge $e$, denoted $|e|$, as the number of nodes incident to $e$. A hypergraph $G = (V, E)$ is *k-uniform* if all edges $e \in E$ have cardinality $|e| = k$. We call a 2-uniform hypergraph a *graph*, and its hyperedges *edges*. We formally define the *hypergraph-alignment problem* as follows.

**Hypergraph Alignment Problem.** Given two hypergraphs $G_s = (V_s, E_s)$ and $G_t = (V_t, E_t)$, with $|V_s| = |V_t|$, the goal is to find a bijective mapping $\varphi : V_s \to V_t$ that maximizes

$$\sum_{v \in V_s} \mathbb{1}[\varphi(v) = \tau(v)], \tag{1}$$

where $\tau : V_s \to V_t$ is the (unknown) ground-truth mapping.

We consider the unsupervised and unrestricted setting, where the alignment must be inferred solely from the structure of the hypergraphs. That is, we do not assume access to node or hyperedge features, side information, or known labels. Moreover, we may assume $|V_s| = |V_t|$ without loss of generality by padding the smaller vertex set with isolated (dummy) nodes.

**Graph representations.** Hypergraphs can be represented as conventional graphs in two ways. The *bipartite representation* encodes a hypergraph $G = (V, E)$ as a bipartite graph[1] $B(G) = (V \cup W, F)$, where $W$ contains a node $w_e$ for each hyperedge $e \in E$, and $(u, w_e) \in F$ if and only if $u \in e$. This encoding is lossless. Second, the *clique representation* builds a graph $C(G)$ by replacing each hyperedge $e$ with a clique on its nodes. This can lead to information loss, as the original hypergraph cannot generally be recovered. While such graph-based representations could, in principle, be used for hypergraph alignment, our experiments (Section 5) show that they are ineffective: the clique view discards structural information, and the bipartite view significantly increases the problem size. See Figure 3 in the Appendix for an illustration of the representations.

### 3.1 GROMOV-WASSERSTEIN DISCREPANCY

Our Gromov-Wasserstein-based framework operates directly on hypergraphs without reducing them to graphs. It combines the ideas of filtration, which is commonly used in persistent homology (Otter et al., 2017), with Gromov-Wasserstein (GW) learning (Peyré et al., 2016; Xu et al., 2019b;a) via the GW discrepancy, which generalizes the GW distance to arbitrary dissimilarity matrices (Mémoli, 2011). The GW discrepancy is defined as follows.

**Definition 1.** The Gromov-Wasserstein discrepancy between two measured dissimilarity matrices $(C_s, \mu_s) \in \mathbb{R}^{|V_s| \times |V_s|} \times \Delta_{|V_s|}$ and $(C_t, \mu_t) \in \mathbb{R}^{|V_t| \times |V_t|} \times \Delta_{|V_t|}$ is defined as

$$\min_{T \in \Pi(\mu_s, \mu_t)} \sum_{i,j,\ell,k} L(C_s[i, k], C_t[j, \ell]) T_{ij} T_{k\ell}, \tag{2}$$

where $\Pi(\mu_s, \mu_t) = \{T \in \mathbb{R}^{|V_s| \times |V_t|}_{\geq 0} \mid T \mathbf{1}_{|V_t|} = \mu_s, T^\top \mathbf{1}_{|V_s|} = \mu_t\}$ and $L$ is an element-wise loss function (Peyré et al., 2016).

In the case of conventional graph alignment (instead of hypergraph alignment), given two graphs $G_x = (V_x, E_x)$ with $x \in \{s, t\}$, the pairs $(C_x, \mu_x) \in \mathbb{R}^{|V_x| \times |V_x|} \times \Delta_{|V_x|}$ represent dissimilarity matrices $C_x = [c_{i'j'}] \in \mathbb{R}^{|V_x| \times |V_x|}$ based on the relational interactions $E_x$, the marginal distributions $\mu_x = [\mu_u] \in \Delta_{|V_x|}$ denote the normalized degree distribution of the nodes. Then $T$ denotes the optimal transport between the nodes $V_s$ and $V_t$ of the two graphs, where $T_{ij}$ represents the probability that node $v_i \in V_s$ corresponds to node $v_j \in V_t$.

---

[1]A graph is *bipartite* if the vertex set $V$ of graph $G = (V, E)$ can be partitioned into two sets $U_1$ and $U_2$ such that for all edges exactly one vertex is in $U_1$ and the other in $U_2$.

# 4 MULTI-SCALE GROMOV-WASSERSTEIN HYPERGRAPH ALIGNMENT

We now introduce our framework that aligns two hypergraphs by combining a filtration-driven multi-scale view with Gromov-Wasserstein (GW) optimal transport. Rather than relying on a single pair-wise cost derived from the full hypergraph, we first construct a nested sequence of subhypergraphs via a filtration, where each level captures structural relationships at a different granularity. Each level then yields its own cost matrix and GW transport plan, reflecting how node-node relationships appear under progressively richer hyperedge information. These per-level transport plans are aggregated into a consensus plan, providing a stable and noise-robust alignment.

## 4.1 HYPERGRAPH FILTRATION

To construct the multi-scale representation, we employ a filtration, a fundamental concept in topological and combinatorial data analysis that constructs a nested sequence of structures capturing how features evolve across multiple scales (Aktas et al., 2019). Here we utilize the nested-space perspective from topological data analysis (without employing homology itself) to obtain nested hierarchies of hypergraphs that allow robust alignment under noisy signals. Real-world hypergraphs often contain hyperedges of varying sizes and densities, reflecting structures at different levels of granularity (Lee et al., 2025). For example, small hyperedges may capture localized interactions, while larger ones represent broader groupings or contextual co-occurrences. By applying filtration based on normalized hyperedge size, we utilize structurally-reliable subgraphs at lower granularity levels, gradually incorporating coarser, and potentially more noisy structures, as the scale increases.

**Definition 2.** Given a hypergraph $G = (V, E)$, a weight function $\omega : E \to \mathbb{R}$, and a scale parameter $r \in \mathbb{R}$, the subhypergraph $F_\omega(G, r)$ is induced by the hyperedges $E' = \{e \in E \mid \omega(e) \leq r\}$.

Varying $r$ generates a sequence of nested subhypergraphs. These are connected via inclusion maps representing the embedding of smaller subhypergraphs into larger ones.

**Lemma 1.** Let $G = (V, E)$ be a hypergraph and $r \leq q \in \mathbb{R}$. Then the inclusion map $\iota_{r,q} : F_\omega(G, r) \hookrightarrow F_\omega(G, q)$ embeds $F_\omega(G, r)$ into $F_\omega(G, q)$, preserving its structure.

This lemma implies that subhypergraphs grow monotonically with increasing $r$, forming a natural filtration $\{F_\omega(G, r)\}_{r \in R}$ for some subset $R \subseteq \mathbb{R}$. Figure 4 in the Appendix shows such a filtration.

Since we assume a finite number of hyperedges, there exists a maximum scale $r_{\max}$ such that $F_\omega(G, r') = F_\omega(G, r_{\max})$ for all $r' \geq r_{\max}$. The structure of $F_\omega(G, r)$ only changes at values of $r$ where new hyperedges are added, i.e., at values in $\{\omega(e) \mid e \in E\}$. We call these values *critical scale parameters*, each corresponding to a structural change in the filtration. By selecting a subset $\mathcal{W} \subseteq \{\omega(e) \mid e \in E\}$, we define a discrete filtration $\{F_\omega(G, \eta_m)\}_{m=1}^{\xi}$, where $\eta_1 < \cdots < \eta_\xi$ are the selected critical values and $\xi = |\mathcal{W}|$. We refer to each $F_\omega(G, \eta_m)$ as *filtration level m*.

Next, we define *filtration-based dissimilarities*. Let $\{F_\omega(G, \eta_m)\}_{m=1}^{\xi}$ be a filtration of $G = (V, E)$. We define the dissimilarities between nodes $u, v \in V$ based on $F_\omega(G, \eta_m)$, which capture the observed dissimilarity at filtration level $m \in [\xi]$. For each $m \in [\xi]$, we define the cost matrix $C^m \in \mathbb{R}^{|V| \times |V|}$ to capture pairwise node dissimilarities based on their co-occurrence in hyperedges from $F_\omega(G, \eta_m)$. Specifically, the entry $C^m[u, v]$ is given by:

$$C^m[u, v] = \frac{1}{\delta^m(u, v) + 1}, \tag{3}$$

where $\delta^m(u, v)$ is the number of hyperedges in $F_\omega(G, \eta_m)$ that contain both nodes $u$ and $v$ in $V$. For diagonal entries, we set $C^m[u, u] = 0$. Thus $C^m[u, v]$ ensures that node pairs sharing more hyperedges at level $m$ have smaller dissimilarities, while pairs with no shared hyperedges receive the maximum dissimilarity of 1. Equation (3) is motivated by the observation that node similarity in hypergraphs is naturally reflected in their co-occurrence within hyperedges (Antelmi et al., 2023). Moreover, in the extremal case where all hyperedges have distinct weight $\omega(e)$, the resulting filtration sequence is in fact lossless.

**Theorem 1.** Let $G = (V, E)$ be a hypergraph in which every hyperedge has size at least 2 and let $\omega : E \to \mathbb{R}$ be a weight function that assigns pairwise distinct values to all hyperedges. Let the filtration be induced by $\omega$ and use all critical thresholds, so that at level $m$ the subhypergraph

contains exactly the edges $e_1, \ldots, e_m$ ordered by increasing weight $\omega(e_1) < \cdots < \omega(e_\xi)$, where $\xi = |E|$. Let $C^m$ be the dissimilarities at level $m$. Then the mapping $G \to \{C^m\}_{m=1}^\xi$ is injective.

In the following, we apply the above construction separately to $G_x = (V_x, E_x)$ for $x \in \{s, t\}$, writing $C_x^m \in \mathbb{R}^{|V_x| \times |V_x|}$ built from $F_\omega(G_x, \eta_m)$.

## 4.2 FILTRATION-BASED MULTI-SCALE GW CONSENSUS

To fully utilize our hypergraph filtration framework, we use the GW discrepancy to integrate multiple pairs of measured dissimilarity matrices, capturing multi-scale structure through globally aligned, mass-preserving transport plans. Now, let $\mathcal{W}_\gamma$ be a set of $\xi$ critical scale parameters (determined by a density parameter $\gamma$; we provide details on how to choose $\mathcal{W}_\gamma$ in Section 4.4). For each $m \in [\xi]$ and $x \in \{s, t\}$, let the per-level dissimilarities $C_x^m$ be as defined in the previous section. We then define a multi-scale GW consensus transport plan as the aggregation of the $\xi$ per-level GW objectives over a shared feasibility region, with independent transport plans at each level.

**Definition 3** (Consensus Coupling). Let $\{C_s^m\}_{m=1}^\xi$ and $\{C_t^m\}_{m=1}^\xi$ be dissimilarity matrices at $\xi$ filtration levels, and let $\mu_s$ and $\mu_t$ be node marginals. For each level $m \in [\xi]$, we compute

$$T^{m\star} := \arg\min_{T^m \in \Pi(\mu_s, \mu_t)} \sum_{i,j,k,\ell} L\big(C_s^m[i,k],\, C_t^m[j,\ell]\big)\, T_{ij}^m\, T_{k\ell}^m,$$

where $\Pi(\mu_s, \mu_t) = \{T \in \mathbb{R}_{\geq 0}^{|V_s| \times |V_t|} \mid T\mathbf{1}_{|V_t|} = \mu_s,\ T^\top \mathbf{1}_{|V_s|} = \mu_t\}$ and $L$ is an element-wise loss (e.g., $L(a, b) = (a - b)^2$). For a probability vector $w = (w_1, \ldots, w_\xi) \in \Delta_\xi$, the consensus coupling is the convex mixture

$$\widehat{T} := \sum_{m=1}^\xi w_m\, T^{m\star} \in \Pi(\mu_s, \mu_t).$$

For the marginal distributions $\mu_x$ with $x \in \{s, t\}$ (that are common for all levels), we use the normalized node degree distribution, where a node's degree is its count of incident hyperedges. In the following, we establish (i) well-posedness and uniqueness of the consensus transport plan (Theorem 2), (ii) its stability to per-level errors (Theorem 3), and (iii) near-optimality of simple uniform weighting under a natural correlated error model (Theorem 4).

**Theorem 2.** The consensus coupling $\widehat{T}$ lies in $\Pi(\mu_s, \mu_t)$ and uniquely minimizes $\min_{T \in \Pi(\mu_s, \mu_t)} \sum_{m=1}^\xi w_m \|T - T^{m\star}\|_F^2$.

Furthermore, the aggregation yields stability across the different scales and small per-level perturbations aggregate linearly as shown in the following.

**Theorem 3.** Suppose that under perturbations of the costs $(C_s^m, C_t^m) \mapsto (\widetilde{C}_s^m, \widetilde{C}_t^m)$, the corresponding optimizers satisfy $\|T^{m\star} - \widetilde{T}^{m\star}\|_F \leq \delta_m$ for all $m$. Then the consensuses obey $\|\widehat{T} - \widetilde{\widehat{T}}\|_F \leq \sum_{m=1}^\xi w_m \delta_m$.

The consensus $\widehat{T}$ provides robustness which stems from two complementary principles: the nature of structural noise in hypergraphs and the stability of the aggregation itself. First, our size-based filtration naturally prioritizes more reliable signals. Smaller hyperedges, which appear early and persist through later filtration levels, have their structural signals reinforced repeatedly. Conversely, very large (and potentially noisy) hyperedges influence fewer levels. This prioritization is motivated by realistic noise models: under random incidence corruption (e.g., flips with probability $p$), the expected number of errors in a hyperedge scales with its cardinality $p|e|$. Thus, larger hyperedges are a priori more likely to be distorted, making the reinforcement of smaller, cohesive structures a robust strategy.

Second, the aggregation step is inherently stable. A simple uniform average ($w_m = 1/\xi$) is a nearly optimal choice in the following setting. Let $T^{m\star} \in \mathbb{R}_+^{n_s \times n_t}$ be the per-scale GW couplings for $m = 1, \ldots, \xi$, and let the consensus be $\widehat{T}_w = \sum_{m=1}^\xi w_m T^{m\star}$ with $w \in \Delta_\xi$. For the unknown target transport $T^\star$, we decompose $T^{m\star} = T^\star + \varepsilon_m$, with $\mathbb{E}[\varepsilon_m] = 0$, under the assumption of zero-mean perturbations capturing finite-sample noise, entropic smoothing, and modeling mismatch.

Under squared Frobenius loss, we define the risk $\mathcal{R}(w) = \mathbb{E}\left[\|\widehat{T}_w - T^\star\|_F^2\right] = w^\top \Sigma w$ where the covariance matrix $\Sigma \in \mathbb{R}^{\xi \times \xi}$ has entries $\Sigma_{mn} = \mathbb{E}\left[\langle \varepsilon_m, \varepsilon_n \rangle_F\right]$. The optimal weights under the linear constraint $\mathbf{1}^\top w = 1$ are then $w^\star \propto \Sigma^{-1}\mathbf{1}$.

**Theorem 4.** If per-scale errors are equicorrelated, i.e., $\Sigma = \sigma^2\left((1-\rho)I + \rho\,\mathbf{1}\mathbf{1}^\top\right)$ with $\rho \in [0,1)$, then $w^\star = \frac{1}{\xi}\mathbf{1}$ ,i.e.,uniform. Moreover, for any $w \in \Delta_\xi$, $\mathcal{R}(w) - \mathcal{R}\left(\frac{1}{\xi}\mathbf{1}\right) = \sigma^2(1-\rho)\left\|w - \frac{1}{\xi}\mathbf{1}\right\|_2^2$.

The equicorrelation model fits our setting as later filtration levels subsume earlier ones, making $T^{m\star}$ highly collinear across $m$; thus off-diagonals of $\Sigma$ are large and roughly homogeneous. Moreover, every $T^{m\star}$ obeys the same $(\mu_s, \mu_t)$ and is smoothed by the same entropy scale, reducing between-level variability beyond signal, thereby justifying the equicorrelation assumption.

Even though uniform weighting is nearly optimal in this equicorrelation model setting, for the general case, we introduce a data-driven weighting scheme. We let the data decide how to weight scales via a leave-one-out agreement. For each level $m$, let $T^{m\star} \in \mathbb{R}^{n_s \times n_t}$ be the optimal coupling and $v_m = \text{vec}(T^{m\star}) \in \mathbb{R}^d$ with $d = n_s n_t$. We define the agreement score

$$s_m = \left\langle T^{m\star}, \overline{T}_{-m} \right\rangle_F, \qquad \text{with} \quad \overline{T}_{-m} = \frac{1}{\xi-1}\sum_{n \neq m} T^{n\star},$$

i.e., how well level $m$ aligns with the consensus of all other levels. We obtain the final weights by a softmax on standardized scores $s_m$, with a hyperparameter $\beta_w$ controlling sharpness. Our weights reward total similarity to other levels while penalizing self-energy. See Appendix B for details.

---

**Algorithm 1: FALCON**

**Input:** Hypergraphs $G_s, G_t$; filtration density $\gamma \in (0,1]$; entropic weight $\beta > 0$
**Output:** Bijective node mapping $\varphi : V_s \to V_t$

1 Select critical scales $\mathcal{W}_\gamma$ and build $\xi = |\mathcal{W}_\gamma|$ filtration levels (Section 4.4)
2 Build marginals $\mu_s, \mu_t$ and per-level costs $\{(C_s^m, C_t^m)\}_{m=1}^\xi$
3 $T^{\text{init}} \leftarrow \mu_s \mu_t^\top$
4 **for** $m = 1$ **to** $\xi$ **do**
5 $\quad\left\lfloor\; T^{m\star} \leftarrow \arg\min_{T \in \Pi(\mu_s, \mu_t)} \mathcal{L}_{\text{GW}}(C_s^m, C_t^m; T) - \beta H(T)$ initialized at $T^{\text{init}}$
6 Determine weights $w = (w_1, \ldots, w_\xi) \in \Delta_\xi$ (uniform or data-driven)
7 Build consensus $\widehat{T} \leftarrow \sum_{m=1}^\xi w_m T^{m\star}$
8 Solve Hungarian on $-\widehat{T}$ to get $\varphi$

---

### 4.3 THE FALCON ALGORITHM

Algorithm 1 shows our principled algorithm FALCON that computes the consensus transport and then decodes a bijective node mapping. Let $\mathcal{W}_\gamma$ denote the set of critical scale parameters selected by density $\gamma$ (see Section 4.4), with $\xi = |\mathcal{W}_\gamma|$ filtration levels. For each level $m \in [\xi]$ and hypergraph $G_x$ with $x \in \{s, t\}$, we build a structural dissimilarity $C_x^m \in \mathbb{R}^{n_x \times n_x}$ from hyperedge co-occurrence counts on the filtered hypergraph $G_x$, and we set node marginals $\mu_x$ as the normalized degree distribution. Given cost pair $(C_s^m, C_t^m)$ and marginals $(\mu_s, \mu_t)$, we solve the standard entropically-regularized GW subproblem

$$T^{m\star} = \arg\min_{T \in \Pi(\mu_s, \mu_t)} \mathcal{L}_{\text{GW}}(C_s^m, C_t^m; T) - \beta H(T), \tag{4}$$

where $H(T) = -\sum_{u,v} T_{uv}(\log T_{uv} - 1)$ is entropic regularization, and $\mathcal{L}_{\text{GW}}$ is Definition 1 instantiated with the squared loss $L(a,b) = (a-b)^2$. We use a KL-proximal/entropic GW solver (Cuturi, 2013; Peyré et al., 2016; 2019) as an off-the-shelf routine.

Algorithm 1 solves Equation (4) independently at each level, initialized with the outer product $T^{\text{init}} = \mu_s \mu_t^\top$. We then determine the weights $w \in \Delta_\xi$, either uniform or data-driven based on the computed $T^{m\star}$, and form the consensus transport plan $\widehat{T} = \sum_{m=1}^\xi w_m T^{m\star}$. Finally, we get the bijective mapping $\varphi : V_s \to V_t$ by solving a linear assignment problem on the similarity $-\widehat{T}$ using the Hungarian method (Kuhn, 1955).

**Theorem 5.** Assume $|V_s| = |V_t| =: n$ and $\xi$ filtration levels. Let $K$ be the number of KL-proximal outer iterations per per-level GW solve. Using dense operations, the time complexity is $\mathcal{O}(\xi \cdot (|E| \cdot n^2 + Kn^3))$, which reduces to $\mathcal{O}(\xi K n^3)$ for $|E| = \mathcal{O}(Kn)$, and the space complexity is $\mathcal{O}(\xi n^2)$ to store $\{C_s^m, C_t^m\}_{m=1}^{\xi}$ and the transport plans.

The complexity of $\mathcal{O}(\xi K n^3)$ is comparable to state-of-the-art graph aligners, such as GWL (Xu et al., 2019b) or FUGAL (Bommakanti et al., 2024).

## 4.4 Scale Parameter Selection

For the filtration, we employ normalized size of hyperedges $\omega_{\text{size}}(e) = |e|/s_{\max}$ with $s_{\max} = \max_{e \in E} |e|$ (see Figure 4 in the appendix for an example). The chosen weights define the filtration, i.e., cumulative subhypergraphs $F_\omega(G, \eta)$ over increasing thresholds $\eta$. For critical-parameter selection, we consider the sets of scale parameters $\mathcal{W}_s$ and $\mathcal{W}_t$ of the two hypergraphs and define $\mathcal{W}_{s \cup t} = \mathcal{W}_s \cup \mathcal{W}_t$. We then choose the subset $\mathcal{W}_\gamma \subseteq \mathcal{W}_{s \cup t}$ of size $c = \lceil \gamma \cdot |\mathcal{W}_{s \cup t}| \rceil$ via a two-sided support rule: sweeping thresholds in ascending order, we place a split whenever both $G_s$ and $G_t$ have accumulated at least one additional hyperedge since the previous split; if fewer than $c$ such points exist, we pad with the largest threshold (see Appendix C for details). Our two-sided support rule ensures that we only compare scales where both hypergraphs have undergone a structural change. This avoids trivial comparisons where one hypergraph's structure is static.

## 5 Experiments

We discuss the following research questions: **RQ1:** How does FALCON compare to state-of-the-art methods under structural perturbations? And how robust is the method to different noise types and levels? **RQ2:** How does the running time compare to the baselines? **RQ3:** How does the hyperparameter $\gamma$ impact the accuracy and the running time?

We provide additional ablation studies on the filtration method and the cost function in Appendix F.

Table 1: Dataset statistics.

| Dataset | $|V|$ | $|E|$ | Min $|e|$ | Max $|e|$ | Avg. $|e|$ | Avg. deg$(u)$ | Domain |
|---------|-------|-------|-----------|-----------|------------|---------------|--------|
| *Pollinator* | 130 | 401 | 2 | 104 | 6.60 | 20.36 | Ecology |
| *NDC-Classes* | 628 | 796 | 2 | 39 | 7.20 | 9.12 | Pharmacology |
| *Email-EU* | 986 | 24 520 | 2 | 40 | 3.62 | 90.04 | Communication |
| *Dawn* | 2 290 | 138 742 | 2 | 16 | 3.99 | 241.55 | Healthcare |

**Datasets:** We benchmark our approach on four real-world hypergraphs covering a range of domains and non-uniform hypergraph characteristics (details in Appendix D). Table 1 shows the dataset statistics. To create alignment tasks, we generate a target hypergraph $G_t$ by systematically perturbing a source hypergraph $G_s$ from each dataset. We apply three challenging types of structural noise: (1) node removals, (2) incidence noise, and (3) hyperedge additions. For each noise type, we use five probability levels (*noise ratio $p$*, from lower to higher noise) to control the perturbation intensity. The node identities in $G_t$ are then randomly permuted to define the ground-truth mapping for evaluation. See Appendix D for details about the datasets and noise types.

**Algorithms and experimental setup:**

Since unsupervised hypergraph alignment lacks established benchmark methods[2], we primarily benchmark against strong unsupervised graph alignment methods. We include the state-of-the-art baselines GWL (Xu et al., 2019b), SGWL (Xu et al., 2019a), REGAL (Heimann et al., 2018), PARROT (Zeng et al., 2023), and FUGAL (Bommakanti et al., 2024). Because these methods are

---

[2]An intuitive baseline for our task would be HyperAlign (Do & Shin, 2024). We ran the authors' public code (GitHub commit 26ae732); the program terminates without producing a non-trivial transport plan, so no alignment accuracy (other than $\approx 0$) can be computed. In direct communication the authors confirmed that their approach and released code suffer from reproducibility issues and they have acknowledged this publicly on the project's repository (https://github.com/manhtuando97/HyperAlign). At the time of writing, they are working on reproducing their own findings or issuing a corrigendum. Consequently, we do not include HyperAlign in our evaluation.

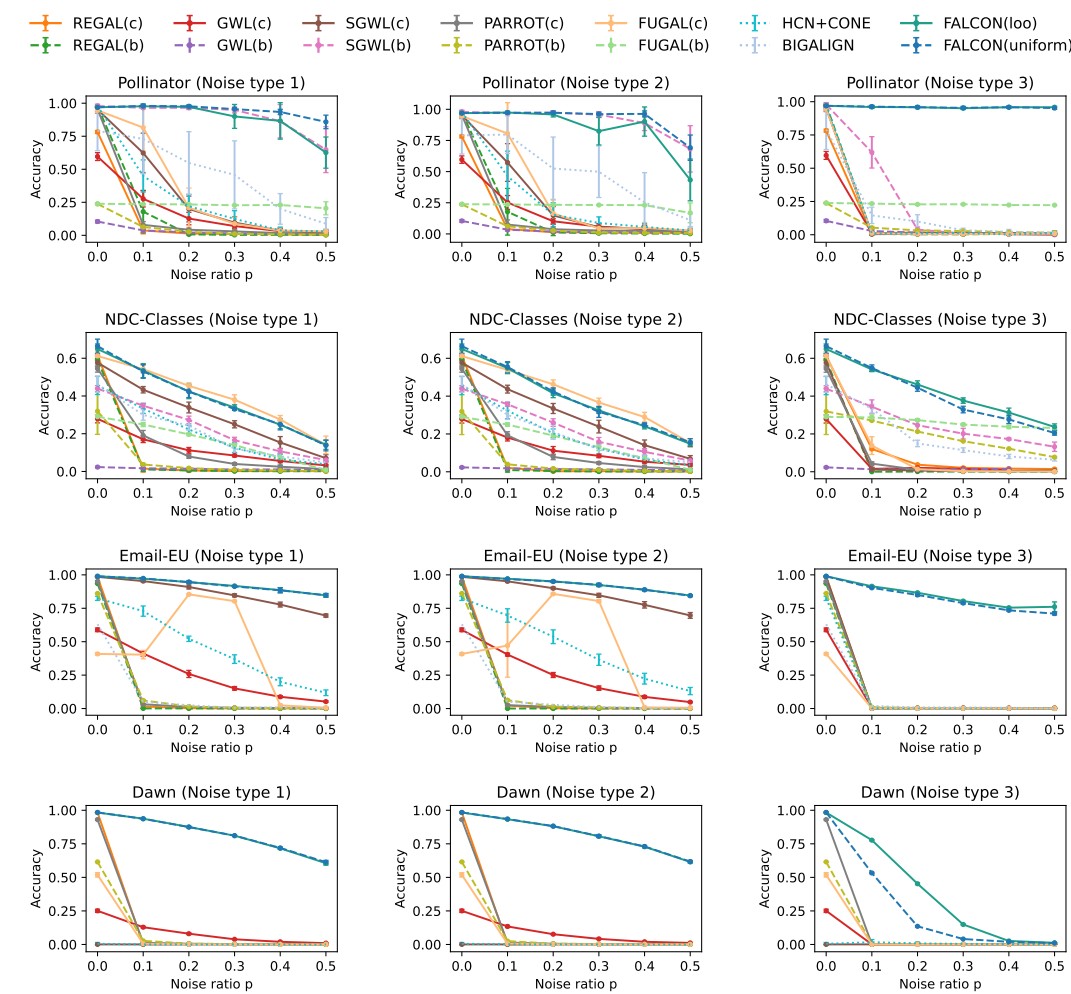

Figure 1: Accuracy results on real-world datasets under different noise types (node removals, incidence flips, hyperedge additions).

Table 2: Average running times (± std.) over all runs (OOT—out of time, OOM—out of memory).

| Algorithm | Pollinator | NDC-Classes | Email-EU | Dawn |
|---|---|---|---|---|
| REGAL(c) | $0.20 \pm 0.15$ | $0.89 \pm 0.19$ | $3.82 \pm 2.03$ | $10.36 \pm 0.44$ |
| REGAL(b) | $0.51 \pm 0.42$ | $1.95 \pm 0.24$ | $73.31 \pm 7.93$ | OOM |
| GWL(c) | $16.59 \pm 1.49$ | $70.01 \pm 7.86$ | $88.20 \pm 4.19$ | $398.40 \pm 27.27$ |
| GWL(b) | $19.98 \pm 11.40$ | $134.28 \pm 11.93$ | OOT | OOM |
| SGWL(c) | $1.02 \pm 0.39$ | $6.58 \pm 3.82$ | $38.45 \pm 6.74$ | $71.77 \pm 2.12$ |
| SGWL(b) | $9.56 \pm 4.24$ | $71.32 \pm 49.17$ | OOT | OOM |
| PARROT(c) | $5.87 \pm 1.99$ | $1.70 \pm 1.10$ | $31.51 \pm 10.70$ | $120.65 \pm 27.82$ |
| FUGAL(c) | $8.95 \pm 2.85$ | $15.30 \pm 6.78$ | $217.49 \pm 384.30$ | $374.14 \pm 310.64$ |
| FUGAL(b) | $10.52 \pm 5.18$ | $115.56 \pm 35.49$ | OOT | OOM |
| HCN+CONE | $4.62 \pm 1.86$ | $0.86 \pm 0.02$ | $1.16 \pm 0.08$ | $3.80 \pm 0.41$ |
| BIGALIGN | $0.47 \pm 0.06$ | $1.19 \pm 0.09$ | $1985.87 \pm 101.32$ | OOM |
| FALCON(loo) | $0.75 \pm 0.02$ | $1.47 \pm 0.18$ | $13.80 \pm 1.63$ | $73.30 \pm 5.50$ |
| FALCON(uniform) | $0.74 \pm 0.08$ | $1.47 \pm 0.18$ | $13.80 \pm 1.62$ | $73.02 \pm 5.61$ |

designed for conventional graphs, we evaluate each using both the clique (c) and bipartite (b) hypergraph representations. We also include BIGALIGN (Koutra et al., 2013), which is specifically designed for bipartite graphs. Finally, as a native hypergraph baseline, we introduce HCN+CONE, which computes hypergraph-aware embeddings via HCN (Bai et al., 2021) and aligns them using the CONE transformation (Chen et al., 2020). See Appendix E for implementation details.

Our algorithm, denoted by FALCON, is implemented in Python 3.9 and PyTorch 2.5.1. We set the hyperparameter $\gamma = 1$ unless stated otherwise. We use two variants: one using uniform weighting (uniform) and one using the data-driven weighting ($\beta_w = 1$) based on leave-one-out (loo). Finally, for the Sinkhorn solver we use $\beta = 0.1$, $K = 200$ outer iterations, and $I_{\text{in}} = 10$ inner iterations. These choices worked well across all experiments and did not require extensive tuning.

All experiments were run on a computer cluster. Each experiment ran exclusively on a node with an Intel(R) Xeon(R) Gold 6130 CPU @ 2.10 GHz, 384 GB of RAM, and an NVIDIA A100 GPU. We used a time limit of one hour. Our source code and the datasets are anonymously available.[3]

### 5.1 RESULTS

**RQ1: Accuracy and robustness.** Figure 1 reports mean accuracy (fraction of correctly matched nodes) over ten runs, with error bars showing standard deviation. Across datasets and noise conditions, both FALCON variants typically achieve the highest or near-highest accuracy. On *NDC-Classes*, FUGAL(c) is often the strongest baseline and even slightly exceeds FALCON at some intermediate noise ratios under noise types 1 and 2, while FALCON remains highly competitive. On *Email-EU*, FUGAL(c) exhibits a non-monotonic accuracy curve as noise increases, which we attribute to strong structural symmetries so moderate noise can sometimes move the solution closer to or further from the reference permutation. On smaller datasets (*Pollinator*, *NDC-Classes*) most methods do well at low noise, but graph-based baselines degrade sharply as perturbations grow, especially on the larger *Email-EU* and *Dawn*. Bipartite variants often fail to complete (scalability; see **RQ2**), while clique-based methods scale but decline steadily. Noise type 3 (hyperedge addition) is particularly challenging: random large hyperedges induce spurious cliques that overwhelm alignments, yet FALCON remains robust. In our size-based filtration, small hyperedges enter early and persist across levels, so any random bi-incidence errors (types 1 and 2) affecting them are inherited by all subsequent levels, creating a strong shared error component across scales. This yields an approximately equicorrelated per-scale error structure, under which Theorem 3 implies uniform averaging is near-optimal; accordingly, FALCON(uniform) and FALCON(loo) perform nearly identically. In contrast, type 3 noise injects large random hyperedges only at late levels, breaking this alignment; leave-one-out weighting down-weights these corrupted levels, so FALCON(loo) outperforms FALCON(uniform). In summary, FALCON 's multi-scale filtration and principled GW consensus yield resilience to structural noise and achieves state-of-the-art accuracy.

**RQ2: Efficiency.** Table 2 reports running times in seconds. Both FALCON variants (uniform, loo) are substantially faster than the strongest accuracy competitors GWL(c) and SGWL(c), with the gap widening on larger datasets. The main scalability bottleneck is the bipartite representation, which expands the node set from $|V|$ to $|V| + |E|$, leading to memory/time failures on large hypergraphs. Clique-based methods keep $|V|$ nodes but operate on dense graphs from clique expansion; by working directly on the native hypergraph, FALCON is more efficient. As the running time linearly depends on $\xi$, we provide an empirical analysis of the number of filtration levels $\xi$ in Appendix F.3, showing that $\xi$ remains small ($< 40$) across all datasets. FALCON uses at most $\approx 1.2$ GiB CPU RAM and 969 MiB GPU VRAM (see Table 4). Although REGAL(c) and PARROT(c) are often faster, they give up substantial accuracy and robustness (see **RQ1**). Overall, FALCON delivers the best balance of accuracy and scalability.

Table 3: Scalability experiment with average running times ($\pm$ std.) over 10 runs.

| $n$ | 4 000 | 6 000 | 8 000 | 10 000 | 12 000 | 14 000 | 16 000 |
|---|---|---|---|---|---|---|---|
| Runtime (s) | $81.9 \pm 0.3$ | $151.3 \pm 0.9$ | $288.1 \pm 1.9$ | $558.7 \pm 3.4$ | $891.6 \pm 5.2$ | $1\,508.3 \pm 6.1$ | $2\,261.8 \pm 9.9$ |

We additionally evaluate scalability on synthetic hypergraphs generated by the following random model. For a chosen number of nodes $n$, we sample $m = 10^5$ hyperedges independently by first choosing an edge size uniformly from the interval $[45, 50]$ and then drawing that many distinct nodes uniformly at random. This produces the source hypergraph $G_s$, which contains all sampled hyperedges. The target hypergraph $G_t$ is obtained by copying $G_s$ and applying a random permutation to its node labels while retaining the ground-truth correspondence. We report the average runtimes

---

[3]https://gitlab.com/anonymous_iclr/falcon

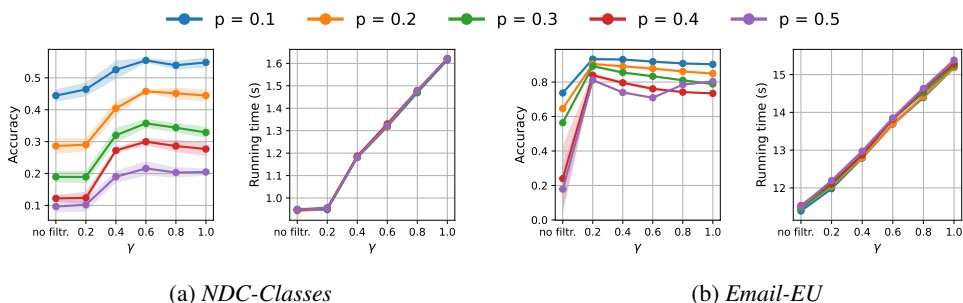

Figure 2: Effect of hyperparameters $\gamma$ on accuracy and running time.

(and standard deviation) of FALCON(uniform) over 10 independent runs in Table 3. The results follow the expected $O(\xi K n^3)$ scaling stated in Theorem 5.

**RQ3: Effect of the hyperparameter $\gamma$.** We evaluate the impact of $\gamma$ on accuracy and running time to assess the robustness of our algorithm to this hyperparameter choice. Figure 2 shows how performance on the *NDC-Classes* and *Email-EU* datasets (see Figure 5 in the appendix for *Pollinator* and *Dawn*) varies with the number of filtration levels, controlled by $\gamma \in \{0.2, \ldots, 1.0\}$, compared to using no filtration (no filtr.). Accuracy generally increases with $\gamma$, as more structural detail is incorporated from a finer-grained filtration. Using no filtration performs significantly worse, confirming the value of our multi-scale approach. For *NDC-Classes*, performance tends to stabilize for $\gamma \geq 0.5$. In some cases we observe a slight performance decrease for higher $\gamma$ when the additional, finer-grained filtration levels introduced are sparse and do not contribute significant new structural information, but instead introduce additional noise into the consensus calculation. The running time increases approximately linearly with $\gamma$, as expected, since more filtration levels require more transport computations. Overall, $\gamma$ controls a clear trade-off between accuracy and computational cost. Importantly, accuracy is stable across a wide range of $\gamma$ values ($\gamma \geq 0.5$), demonstrating that FALCON is robust to this hyperparameter and does not require extensive tuning.

## 6 Conclusion, Limitations, and Future Work

We studied unsupervised hypergraph alignment. Our proposed FALCON algorithm leverages a hypergraph filtration to build multi-scale structural costs and aggregates per-level Gromov-Wasserstein solutions into a stable consensus transport. The empirical results show that FALCON outperforms strong baselines, particularly under noisy perturbations, while maintaining efficient runtime.

**Limitations.** FALCON is designed for non-uniform hypergraphs; in $k$-uniform settings (including graphs) the size-based filtration collapses to a single level, reducing the method to a standard single-scale GW aligner. Furthermore, although our cost matrices incorporate higher-order information by aggregating shared hyperedges, the optimization ultimately relies on pairwise dissimilarity matrices, which are, in general, not lossless. While, e.g., explicit tensor-based matching could offer greater expressiveness, it would be computationally prohibitive in case of large hyperedge sizes; FALCON accepts this trade-off to ensure tractability. However, our approach inherits the cubic time complexity of dense GW solvers, which limits scalability on very large hypergraphs. Finally, our theoretical results focus on the stability and aggregation properties of the multi-scale consensus rather than on exact recovery, which is intractable in general noisy, non-isomorphic settings.

**Future Work.** While our size-based filtration is effective for general hypergraphs, it collapses to a single scale for $k$-uniform ones, motivating the need for alternative criteria for these regular cases. More broadly, the filtration function offers a powerful way to inject domain knowledge or partial supervision into the alignment process; exploring functions based on node centrality or domain-specific attributes is a promising direction. To address the computational bottleneck on very large datasets, replacing the dense GW solver with scalable OT approximations (e.g., low-rank or partition-based solvers) is a natural extension of our modular framework.

## REPRODUCIBILITY AND ETHICAL STATEMENT

**Reproducibility:** To ensure the reproducibility of our work, we provide our source code, datasets, and experiment scripts in an anonymous repository: `https://gitlab.com/anonymous_iclr/falcon`. The core algorithm is detailed in Algorithm 1. The experimental setup, including the computational environment, specific software versions (Python 3.9, PyTorch 2.5.1), and hyperparameters for our method and all baselines, is described in Section 5 and Appendix E. The datasets used are publicly available (links are in the repository), and the procedures for generating the perturbed hypergraphs for alignment tasks are fully specified in Appendix D and the code for generating perturbations is also in our repository.

**Ethical Statement:** Our work concerns structure-only alignment of (hyper)graphs and shares the standard risk profile of network alignment methods, e.g., potential misuse for de-anonymization or sensitive linkage. We mitigate this by (i) using only public benchmark or synthetic datasets; (ii) releasing code with an Intended Use & Restrictions notice that prohibits re-identification and linkage of personal data; and (iii) providing examples and defaults that operate solely on public/synthetic data. We collected no new human-subjects data, and no personal information was processed. All datasets are used under their respective licenses, and we include dataset links, citations, and license pointers in the repository. We report runtime and hardware to support energy-aware replication and efficient re-use. We have read and adhere to the ICLR Code of Ethics.

**GenAI Usage:** AI tools were used for editing and polishing purposes. Specifically, LLMs were employed for light editing tasks such as grammar checking, typo correction, and other minor revisions of author-written text.

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

## APPENDIX

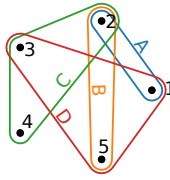
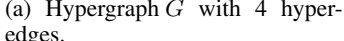
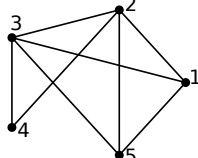
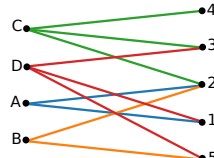

(a) Hypergraph $G$ with 4 hyperedges.

(b) The clique representation of $G$.

(c) The bipartite representation of $G$.

Figure 3: Example of a hypergraph and its representations as conventional graphs.

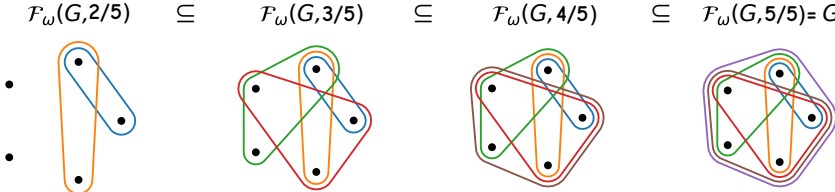

Figure 4: Hypergraph filtration using hyperedge normalized size with $\omega_{\text{size}}(e) = \frac{|e|}{s_{\max}}$ and $s_{\max} = \max_{e' \in E} |e'|$. Here, $s_{\max} = 5$.

# A  OMITTED PROOFS

*Proof of Theorem 1.* Since $C^m(u,v) = 1/(\delta^m(u,v)+1)$, the sequence $\{C^m\}_{m=1}^{\xi}$ is equivalent to the sequence of co-occurrence matrices $\{\delta^m\}_{m=1}^{\xi}$. At level $m$, $\delta^m(u,v)$ counts how many of $e_1, \ldots, e_m$ contain both $u$ and $v$. We reconstruct $E$ by induction on $m$. For $m=1$, the pairs with $\delta^1(u,v) = 1$ form exactly the clique on the node set of $e_1$, so $e_1$ is uniquely determined. Assume $e_1, \ldots, e_{m-1}$ are known. Define $\Delta^m(u,v) := \delta^m(u,v) - \delta^{m-1}(u,v)$. The only new edge between levels $m-1$ and $m$ is $e_m$, hence $\Delta^m(u,v) = 1$ if and only if $u,v \in e_m$ and 0 otherwise. Thus $e_m = \{u \in V : \exists v \in V \text{ with } \Delta^m(u,v) = 1\}$, i.e., the set of all nodes appearing in at least one pair with $\Delta^m(u,v) = 1$ recovers $e_m$ exactly. By induction, all hyperedges are uniquely recovered, so the mapping $G \to \{C^m\}_{m=1}^{\xi}$ is injective. □

*Proof of Theorem 2.* Convexity of $\Pi(\mu_s, \mu_t)$ implies any convex mixture of feasible couplings is feasible, hence $\widehat{T} \in \Pi(\mu_s, \mu_t)$. The objective is strictly convex in $T$; its unconstrained minimizer is $\sum_m w_m T^{m\star}$, which is feasible by convexity, so it is also the constrained minimizer. Uniqueness follows from strict convexity. □

*Proof of Theorem 3.* By linearity and the triangle inequality, $\|\widehat{T} - \widetilde{\widehat{T}}\|_F = \|\sum_m w_m(T^{m\star} - \widetilde{T}^{m\star})\|_F \le \sum_m w_m \|T^{m\star} - \widetilde{T}^{m\star}\|_F \le \sum_m w_m \delta_m$. □

*Proof of Theorem 4.* Write $\mathbf{1} \in \mathbb{R}^{\xi}$ for the all-ones vector and set $\bar{w} := \frac{1}{\xi}\mathbf{1}$. For any feasible $w \in \Delta_{\xi}$ (i.e., $\mathbf{1}^{\top}w = 1$, $w \ge 0$), decompose $w = \bar{w} + u$ with $\mathbf{1}^{\top}u = 0$.

For the equicorrelation matrix,

$$\Sigma = \sigma^2\big((1-\rho)I + \rho\,\mathbf{1}\mathbf{1}^{\top}\big),$$

the quadratic form simplifies for any $w$ to

$$w^{\top}\Sigma w = \sigma^2\big((1-\rho)\|w\|_2^2 + \rho\,(\mathbf{1}^{\top}w)^2\big).$$

Since $\mathbf{1}^{\top}w = 1$ for all feasible $w$, the risk reduces to

$$\mathcal{R}(w) = \sigma^2\big((1-\rho)\|w\|_2^2 + \rho\big).$$

Using $w = \bar{w} + u$ with $\mathbf{1}^{\top}u = 0$ gives $\bar{w}^{\top}u = (1/\xi)\,\mathbf{1}^{\top}u = 0$, hence

$$\|w\|_2^2 = \|\bar{w}\|_2^2 + \|u\|_2^2 \quad \text{with} \quad \|\bar{w}\|_2^2 = \left\|\tfrac{1}{\xi}\mathbf{1}\right\|_2^2 = \tfrac{1}{\xi}.$$

Therefore

$$\mathcal{R}(w) = \sigma^2\Big((1-\rho)\Big(\tfrac{1}{\xi} + \|u\|_2^2\Big) + \rho\Big) = \underbrace{\sigma^2\Big((1-\rho)\tfrac{1}{\xi} + \rho\Big)}_{\mathcal{R}(\bar{w})} + \sigma^2(1-\rho)\|u\|_2^2.$$

Because $\rho < 1$, the coefficient $(1-\rho) > 0$, so $\mathcal{R}(w)$ is strictly minimized when $\|u\|_2 = 0$, i.e., when $w = \bar{w} = \frac{1}{\xi}\mathbf{1}$. This $w^{\star}$ is feasible ($w^{\star} \ge 0$ and $\mathbf{1}^{\top}w^{\star} = 1$), hence it is the unique minimizer over $\Delta_{\xi}$.

For the excess risk, note that $u = w - \bar{w}$, so

$$\mathcal{R}(w) - \mathcal{R}(\bar{w}) = \sigma^2(1-\rho)\|u\|_2^2 = \sigma^2(1-\rho)\left\|w - \tfrac{1}{\xi}\mathbf{1}\right\|_2^2.$$

This completes the proof. □

*Proof of Theorem 5.* Forming $\{C_s^m, C_t^m\}_{m=1}^{\xi}$ from hyperedge co-occurrence counts requires examining all node pairs within each hyperedge at each filtration level. For a hyperedge $e$ with $|e|$ nodes, there are $\binom{|e|}{2} = \mathcal{O}(|e|^2)$ pairs to process. Across all hyperedges in a filtration level, this costs $\mathcal{O}(\sum_{e \in E} |e|^2)$ per level. In the worst case where hyperedges have size $\mathcal{O}(n)$, this becomes $\mathcal{O}(|E| \cdot n^2)$ per level.

For all $\xi$ filtration levels, the cost matrix construction requires $\mathcal{O}(\xi|E|n^2)$. For typical sparse hypergraphs where $|E| = \mathcal{O}(n)$, this simplifies to $\mathcal{O}(\xi n^3)$. When $|E| = \mathcal{O}(Kn)$, we have $\xi|E|n^2 = \mathcal{O}(\xi Kn^3)$, which means the GW solves (analyzed below) dominate the overall complexity.

We solve the entropically-regularized GW problem Equation (4) for each of the $\xi$ levels. We use a standard proximal point solver, which involves $K$ outer iterations (Peyré et al., 2016).Let $I_{\text{in}}$ the number of Sinkhorn scalings per outer iteration. Each outer iteration requires solving an entropic optimal transport (OT) problem. The main cost of this step is constructing the cost matrix for the OT problem, which is derived from the gradient of the GW loss at the current transport plan $T_k$. The computationally dominant term is $-2C_s^m T_k (C_t^m)^T$ (Peyré et al., 2019). This dense matrix multiplication has a complexity of $\mathcal{O}(n^3)$. Once the $n \times n$ cost matrix is formed, the inner OT problem is solved using $I_{\text{in}}$ iterations of the Sinkhorn algorithm, with each iteration costing $\mathcal{O}(n^2)$. Therefore, the complexity of one per-level GW solve is $\mathcal{O}(K(n^3 + I_{\text{in}}n^2))$. Since $I_{\text{in}}$ is typically small, this simplifies to $\mathcal{O}(Kn^3)$. For all $\xi$ levels, the total complexity for the GW solves is $\mathcal{O}(\xi Kn^3)$.

Building the consensus transport plan $\widehat{T}$ by taking a weighted average of the $\xi$ plans $\{T^{m,*}\}_{m=1}^{\xi}$ costs $\mathcal{O}(\xi n^2)$. Solving the linear assignment problem on the final $n \times n$ similarity matrix $-\widehat{T}$ using the Hungarian method costs $\mathcal{O}(n^3)$. Adding these components, the total time complexity is $\mathcal{O}(\xi n^2 + \xi Kn^3 + \xi n^2 + n^3) = \mathcal{O}(\xi Kn^3)$.

The space complexity is dominated by storing the $\xi$ pairs of $n \times n$ cost matrices and the $\xi$ transport plans, which requires $\mathcal{O}(\xi n^2)$ space. $\square$

## B  DATA-DRIVEN WEIGHTING

To emphasize the most informative filtration levels, we let the data decide how to weight scales via a leave-one-out agreement. For each level $m$, let $T^{m\star} \in \mathbb{R}^{n_s \times n_t}$ be the optimal coupling and $v_m = \text{vec}(T^{m\star}) \in \mathbb{R}^d$ with $d = n_s n_t$. We define the agreement score

$$s_m = \left\langle T^{m\star}, \overline{T}_{-m} \right\rangle_F, \qquad \overline{T}_{-m} = \frac{1}{\xi-1} \sum_{n \neq m} T^{n\star},$$

i.e., how well level $m$ aligns (in Frobenius inner product) with the consensus of all other levels. We obtain the final weights by a temperature-scaled softmax on standardized scores $s_m$, with a hyperparameter $\beta_w$ controlling sharpness. Our weights reward total similarity to other levels while penalizing self-energy. Moreover, the consensus remains the weighted Fréchet mean in the transport polytope (Theorem 2), and hence inherits the stability bound of Theorem 3.

Let $G \in \mathbb{R}^{\xi \times \xi}$ be the following Gram matrix in coupling space, $G_{mn} = \langle v_m, v_n \rangle = \langle T^{m\star}, T^{n\star} \rangle_F$. Rearranging gives

$$s_m = \frac{1}{\xi-1}\big((G\mathbf{1})_m - G_{mm}\big),$$

showing that agreement rewards total similarity to other scales (row-sum $(G\mathbf{1})_m$) while penalizing self-energy $G_{mm} = \|T^{m\star}\|_F^2$. Note that our implementation uses the leave-one-out formula directly and does not require forming $G$.

Moreover, we convert scores to weights with a temperature-scaled softmax on standardized scores,

$$\tilde{s}_m = \frac{s_m - \bar{s}}{\text{std}(s)}, \qquad w_m = \frac{\exp(\beta_w \tilde{s}_m)}{\sum_n \exp(\beta_w \tilde{s}_n)}.$$

Standardization makes $\beta_w$ comparable across datasets; $\beta_w$ controls sharpness (small $\beta_w$: near-uniform; large $\beta_w$: concentrate on the highest-agreement levels). This softmax is the closed-form solution of the entropy-regularized linear objective $\max_{w \in \Delta} \beta_w s^\top w + H(w)$ with $H(w) = -\sum_m w_m \log w_m$.

In our size-based filtration, genuinely persistent small-scale structure appears across many levels, whereas noisy large hyperedges appear late and at few levels. Our agreement weighting naturally amplifies multi-level corroboration and down-weights idiosyncratic outliers, yielding a consensus coupling that remains the weighted Fréchet mean inside the feasible transport polytope.

## C    CRITICAL SCALE PARAMETER DETAILS

We provide details on how the critical scale parameters are chosen. For two hypergraphs $G_s$ and $G_t$, let the critical scales be the distinct score values at which the filtrations $F_\omega(G_s, \eta)$ and $F_\omega(G_t, \eta)$ change, i.e.,

$$\mathcal{W}_s = \{\omega_s(e) : e \in E_s\}, \qquad \mathcal{W}_t = \{\omega_t(e) : e \in E_t\}, \qquad \mathcal{W}_{s \cup t} = \mathcal{W}_s \cup \mathcal{W}_t.$$

Let $t = |\mathcal{W}_{s \cup t}|$ and fix a budget $c = \lceil \gamma \cdot t \rceil$ with $\gamma \in (0, 1]$. We then select a subset $\mathcal{W}_\gamma \subseteq \mathcal{W}_{s \cup t}$ of size at most $c$ by the following greedy rule:

1. Sort $\mathcal{W}_{s \cup t}$ increasingly and denote the ordered candidates by $\{\eta^{(1)} < \eta^{(2)} < \cdots < \eta^{(t)}\}$.

2. Sweep through this list and place a split at $\eta^{(k)}$ whenever both $G_s$ and $G_t$ have accumulated at least one additional hyperedge since the last selected split.

3. Continue until $c$ splits have been placed. If fewer than $c$ splits are found, pad with the largest available threshold so that $|\mathcal{W}_\gamma| = c$.

This procedure ensures that selected scales have *two-sided support*, i.e., they correspond to thresholds at which both graphs undergo a non-trivial change. At the same time, the parameter $\gamma$ controls the total number of retained scales: $\gamma = 1$ yields all possible scales, while smaller $\gamma$ subsamples to a coarser set of levels. This avoids thresholds that are empty or nearly empty on one side, while still maintaining comparability across graphs.

## D    DATASETS

We use four real-world hypergraphs from different domains:

- *Pollinator:* This dataset represents a hypergraph where the nodes correspond to plant species, and the hyperedges represent pollinator species that visit each plant. The data is provided by https://www.web-of-life.es/.

- *NDC-Classes:* In this dataset each hyperedge corresponds to a drug and the nodes are class labels assigned to it. The data originate from the National Drug Code (NDC) Directory, released by the U.S. Food and Drug Administration under the Drug Listing Act of 1972. (Benson et al., 2018).

- *Email-EU:* This dataset is a hypergraph of email exchanges at a European research institution, where nodes represent email addresses and each hyperedge corresponds to a message sent to multiple recipients (Yin et al., 2017).

- *Dawn:* In the dataset, nodes correspond to drugs, hyperedges capture the sets of drugs taken by a patient prior to an emergency room visit (Amburg et al., 2020).

To obtain pairs of hypergraphs, $G_s$ and $G_t$, for alignment, we use two instances of each dataset, where the second instance, $G_t$, is a perturbed version of the first. We introduce the following types of noise:

- **Noise type 1 – Node removal:** We randomly remove nodes in $G_t$ from hyperedges with probability $p$.

- **Noise type 2 – Incidence noise:** We randomly flip up to $\sum_{i,j} I_{ij}$ bits in the incidence matrix of $G_t$ with probability $p$.

- **Noise type 3 – Hyperedge addition:** We introduce $k = \lfloor 0.1 \cdot p \cdot |E_s| \rfloor$ new hyperedges to $G_t$, where each new edge's size is set to size $s_{\max} = \max_{e \in E_s} |e|$ and is populated by nodes randomly sampled from $V_t$.

For all three noise types, we use $p \in \{0.1, 0.2, 0.3, 0.4, 0.5\}$.

As is standard in alignment tasks, we randomly permute the node IDs of $G_t$ to define the ground-truth mapping $\tau$. For each hypergraph dataset and noise type, we generate ten independent alignment instances. Although optimal structural mappings can be non-unique, especially in symmetric hypergraphs, we follow common practice and consistently evaluate accuracy against $\tau$ across methods and runs.

Table 4: Peak memory usage of FALCON in Mebibyte (MiB).

| Type | Pollinator | NDC-Classes | Email-EU | Dawn |
|------|-----------|-------------|----------|------|
| Peak RAM usage on the CPU | 782 | 778 | 843 | 1203 |
| Peak VRAM usage on the GPU | 559 | 661 | 981 | 969 |

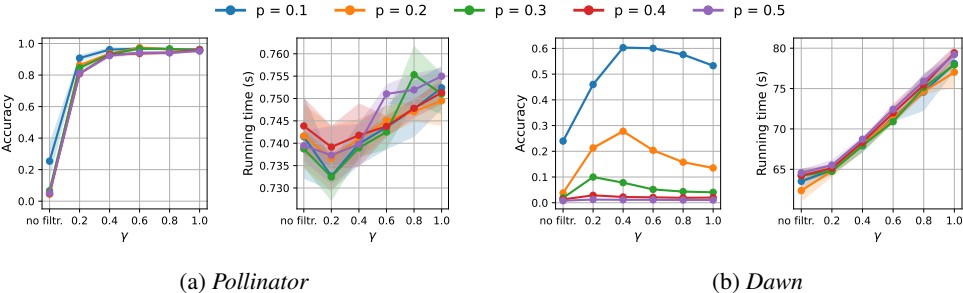

(a) *Pollinator*          (b) *Dawn*

Figure 5: Effect of hyperparameters $\gamma$ on accuracy and running time.

# E   DETAILS ON THE BASELINES

**HCN+CONE details:** The HCN+CONE baseline is a two-stage unsupervised hypergraph alignment method.

*1. Node Embedding (HCN Autoencoder):* Each hypergraph is independently encoded using a two-layer HypergraphConv autoencoder (Bai et al., 2021), with batch normalization and ReLU after the first layer. Key hyperparameters (embedding dim. 64, hidden dim. 128, learning rate 0.01, epochs 512) were selected empirically based on performance.

*2. Embedding Alignment and Matching (CONE Procedure):* L2-normalized embeddings are aligned via the iterative Optimal Transport method from CONE (Chen et al., 2020), and matched using the Hungarian algorithm. We use the standard parameters from (Chen et al., 2020), which we found to perform best in our setting.

**Hyperparameter settings for graph-based baselines:**

- GWL:[1] We used five epochs and set $\beta = 0.1$, the number of outer iterations to $M = 400$, and inner iterations to $N = 100$. And, we used a batch size of $10^6$, as smaller batch sizes resulted in significantly worse performance. These settings were determined empirically and outperformed the default parameters.
- SGWL:[1] We set $\beta = 0.1$, the number of outer iterations to $M = 2000$, inner iterations to $N = 2$, and the number of partition levels to 3. These settings were determined empirically and outperformed the default parameters.
- REGAL:[2] We used the default parameters.
- PARROT:[1] We used the default parameters.
- FUGAL:[3] We used the default parameters.
- BIGALIGN: We implemented the efficient BIGALIGN-SKIP variant presented in (Koutra et al., 2013) in Python. We used the default parameters suggested in (Koutra et al., 2013).

# F   ADDITIONAL EXPERIMENTS

In this section, we provide additional experimental results.

---

[1] https://github.com/constantinosskitsas/Framework_GraphAlignment.
[2] https://github.com/GemsLab/REGAL.
[3] https://github.com/idea-iitd/Fugal

## F.1  IMPACT OF THE DISSIMILARITY MEASURE

We compare our default node–node dissimilarity from Equation (3) with two standard choices. Let $\delta(u, v)$ denote the co-occurrence count, and $\deg(u)$ the per-node degree (number of incident hyper-edges). We define:

$$\textbf{Jaccard:} \quad C_{\text{jac}}(u, v) \;=\; 1 \;-\; \frac{\delta(u, v)}{\deg(u) + \deg(v) - \delta(u, v)}.$$

$$\textbf{Cosine:} \quad C_{\cos}(u, v) \;=\; 1 \;-\; \frac{\delta(u, v)}{\sqrt{\deg(u)\ \deg(v)}}.$$

We evaluate on all datasets and noise types using noise ratio $p = 0.5$. Table 5 shows mean accuracy and running times over ten independent runs using FALCON(uniform). Default denotes Equation (3) and yields the highest accuracy in almost all cases. Only for *Dawn* under noise type 3 do Jaccard and cosine dissimilarities achieve better accuracy.

Table 5: Mean accuracy and standard deviations (best in **bold**).

| **Dissimilarity** | Noise type | *Pollinator* | *NDC-Classes* | *Email-EU* | *Dawn* |
|---|---|---|---|---|---|
| Jaccard | 1 | $0.31 \pm 0.11$ | $0.08 \pm 0.01$ | $0.13 \pm 0.02$ | $0.09 \pm 0.01$ |
| | 2 | $0.28 \pm 0.04$ | $0.09 \pm 0.02$ | $0.12 \pm 0.01$ | $0.09 \pm 0.00$ |
| | 3 | $0.62 \pm 0.02$ | $0.11 \pm 0.01$ | $0.29 \pm 0.01$ | $0.13 \pm 0.00$ |
| Cosine | 1 | $0.45 \pm 0.11$ | $0.10 \pm 0.02$ | $0.22 \pm 0.02$ | $0.19 \pm 0.00$ |
| | 2 | $0.35 \pm 0.03$ | $0.10 \pm 0.03$ | $0.21 \pm 0.02$ | $0.20 \pm 0.01$ |
| | 3 | $0.87 \pm 0.00$ | $0.10 \pm 0.01$ | $0.39 \pm 0.01$ | $\textbf{0.24} \pm 0.01$ |
| Default (Equation (3)) | 1 | $\textbf{0.74} \pm 0.15$ | $\textbf{0.14} \pm 0.02$ | $\textbf{0.85} \pm 0.01$ | $\textbf{0.61} \pm 0.01$ |
| | 2 | $\textbf{0.56} \pm 0.19$ | $\textbf{0.15} \pm 0.02$ | $\textbf{0.84} \pm 0.01$ | $\textbf{0.62} \pm 0.01$ |
| | 3 | $\textbf{0.96} \pm 0.01$ | $\textbf{0.22} \pm 0.02$ | $\textbf{0.74} \pm 0.04$ | $0.01 \pm 0.00$ |

## F.2  IMPACT OF THE FILTRATION WEIGHTS

So far we used for the filtration the normalized size of hyperedges $\omega_{\text{size}}(e) = |e|/s_{\max}$ with $s_{\max} = \max_{e \in E} |e|$. As an alternative to the size-based filtration, we use weights based on min-max normalized average node degree of hyperedges:

$$\deg(v) = \big|\{e \in H : v \in e\}\big|, \qquad \omega_{\text{deg}}(e) = \text{norm}_{\min \max}\left(\frac{1}{|e|} \sum_{v \in e} \deg(v)\right)$$

The idea is to emphasize hyperedges incident to high-degree (central) nodes, so earlier filtration levels prioritize interactions that concentrate network activity rather than large set size. Min-max normalization makes these scores comparable across datasets, providing a scale-free alternative to size reflecting local participation intensity and hub structure.

Figure 6 shows the results for *Pollinator* and *NDC-Classes* where FALCON(uniform, size) uses $\omega_{\text{size}}$ and FALCON(uniform, degree) $\omega_{\text{deg}}$. The size-based filtration leads to significantly better results for all noise types for *Pollinator*. In the case of *NDC-Classes*, for noise types 1 and 2 the filtrations have comparable accuracy. However, for noise type 3, the size-based weighting has a clear advantage. Unlike $\omega_{\text{size}}$, the degree-based $\omega_{\text{deg}}$ does not lead to a natural hierarchy over hyperedges, and averaging incident degree seems to yield diffuse filtration levels and weakens the multi-scale signal.

For the other datasets, *Email-EU* and *Dawn*, FALCON(uniform, degree) run out of memory. The reason is the large number of filtration levels as $\omega_{\text{deg}}$ induces many distinct critical values, so the number of levels $\xi$ can be substantially larger than under $\omega_{\text{size}}$. For example, *Dawn* can have more than $45\,000$ filtration levels using $\omega_{\text{deg}}$ and at most 16 using $\omega_{\text{size}}$.

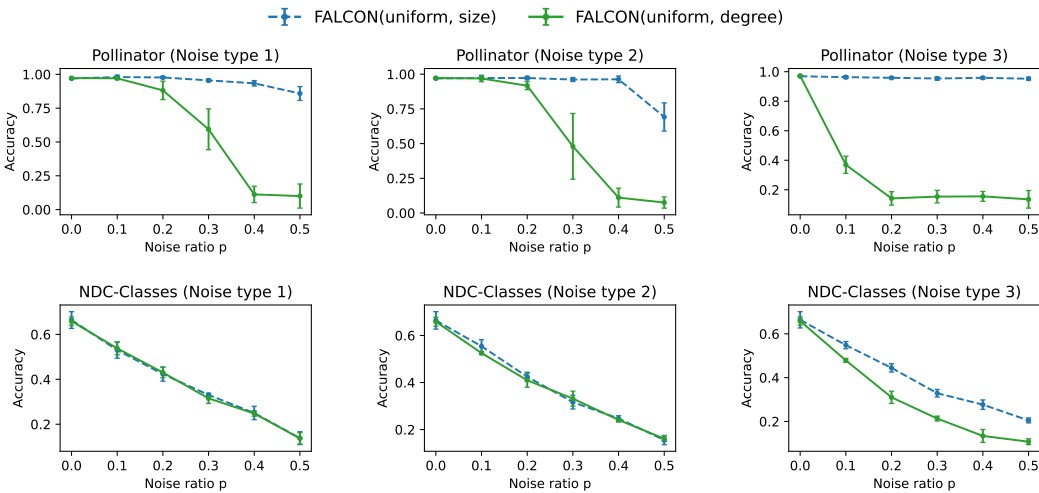

Figure 6: Accuracy results using different filtration weights under different noise types (node removals, incidence flips, hyperedge additions).

### F.3 NUMBER OF FILTRATION LEVELS

We report the average values of $\xi$ in Table 6. As $\xi$ is strictly upper-bounded by the number of distinct hyperedge sizes in the data, it remains consistently small ($< 40$) across all datasets and noise types.

Table 6: Average number of filtration levels $\xi$. The row *Distinct Sizes* denotes the number of unique hyperedge sizes in the unperturbed source hypergraph.

| **Metric** | *Pollinator* | *NDC-Classes* | *Email-EU* | *Dawn* |
|---|---|---|---|---|
| Distinct Hyperedge Sizes | 29 | 25 | 38 | 15 |
| $\xi$ (Noise Type 1) | 22.9 | 18.5 | 30.9 | 13.8 |
| $\xi$ (Noise Type 2) | 22.7 | 19.2 | 30.9 | 13.7 |
| $\xi$ (Noise Type 3) | 29.0 | 25.0 | 38.0 | 15.0 |

Table 7: Notation and symbols.

| Symbol | Meaning |
|---|---|
| $G = (V, E)$ | Hypergraph with node set $V$ and hyperedge set $E$. |
| $|e|$ | Cardinality of hyperedge $e$. |
| $G_x = (V_x, E_x)$ | For $x \in \{s, t\}$, the source and target hypergraphs. |
| $\varphi : V_s \to V_t$ | Bijective node mapping. |
| $[k] = \{1, \dots, k\}$ | Index set shorthand. |
| $\omega : E \to \mathbb{R}$ | Hyperedge weight used for filtration. |
| $F_\omega(G, r)$ | Subhypergraph induced by $\{e \in E : \omega(e) \leq r\}$ at scale $r$. |
| $\omega_{\text{size}}(e) = |e|/s_{\max}$ | Normalized size weight with $s_{\max} = \max_{e \in E} |e|$. |
| $\mathcal{W}_s, \mathcal{W}_t$ | Sets of critical scales for $G_s$ and $G_t$. |
| $\mathcal{W}_{s \cup t} = \mathcal{W}_s \cup \mathcal{W}_t$ | Union of critical scales (two-sided support). |
| $\mathcal{W}_\gamma \subseteq \mathcal{W}_{s \cup t}$ | Selected scale subset controlled by density $\gamma \in (0, 1]$. |
| $\xi$ | Number of filtration levels/selected scale parameter, e.g., $\xi = |\mathcal{W}_\gamma|$. |
| $\{\eta_m\}_{m=1}^\xi$ | Ordered critical thresholds (filtration levels). |
| $\delta^m(u, v)$ | Co-occurrence count of nodes $u, v$ in $F_\omega(G, \eta_m)$. |
| $\Delta_k$ | Probability simplex of dimension $k \in \mathbb{N}$. |
| $C_x^m \in \mathbb{R}^{|V_x| \times |V_x|}$ | Dissimilarity matrix at level $m$ for $G_x$. |
| $\mu_x \in \Delta_{|V_x|}$ | Node marginal on $V_x$ (normalized degree distribution). |
| $L(\cdot, \cdot)$ | Element-wise loss in GW (e.g., $(a - b)^2$). |
| $\langle A, B \rangle_F = \text{tr}(A^\top B)$ | Frobenius inner product |
| $\Pi(\mu_s, \mu_t)$ | Feasible couplings $\{T \geq 0 : T\mathbf{1} = \mu_s, \ T^\top \mathbf{1} = \mu_t\}$. |
| $T^{m\star}$ | Per-level optimal GW coupling at level $m$. |
| $\widehat{T} = \sum_{m=1}^\xi w_m T^{m\star}$ | Consensus coupling (weighted Fréchet mean in coupling space). |
| $w \in \Delta_\xi$ | Weight vector $w \in \mathbb{R}_{\geq 0}^\xi$ with $\sum_m w_m = 1$ for consensus building. |
| $\beta \in \mathbb{R}_{>0}$ | Entropic regularization strength in per-level solves. |
| $H(T)$ | Entropy of transport plan $T$. |
| $T^\star$ | Unknown optimal target transport. |
| $\varepsilon_m = T^{m\star} - T^\star$ | Per-level error. |
| $\widehat{T}_w = \sum_{m=1}^\xi w_m T^{m\star}$ | Consensus estimator for given $w$. |
| $\mathcal{R}(w) = \mathbb{E}\left[\|\widehat{T}_w - T^\star\|_F^2\right]$ | Mean-squared risk under Frobenius loss. |
| $\Sigma \in \mathbb{R}^{\xi \times \xi}$ | Covariance across scales; $\Sigma_{mn} = \mathbb{E}[\langle \varepsilon_m, \varepsilon_n \rangle_F]$. |
| $\mathbf{1} \in \mathbb{R}^\xi$ | All-ones vector of dimension $\xi$. |
| $w^\star = \dfrac{\Sigma^{-1} \mathbf{1}}{\mathbf{1}^\top \Sigma^{-1} \mathbf{1}}$ | Minimum-variance weights under $\mathbf{1}^\top w = 1$. |
| $\sigma^2, \rho$ | Variance and correlation in $\Sigma = \sigma^2((1 - \rho)I + \rho \mathbf{1}\mathbf{1}^\top)$. |
| $s_m = \langle T^{m\star}, \overline{T}_{-m} \rangle_F$ | Leave-one-out agreement score. |
| $\overline{T}_{-m} = \frac{1}{\xi-1} \sum_{n \neq m} T^{n\star}$ | Average coupling excluding level $m$. |
| $\beta_w \in \mathbb{R}_{>0}$ | Softmax sharpness for data-driven weights. |
| $n_s = |V_s|, \ n_t = |V_t|, \ n$ | Numbers of nodes; sometimes $n := |V_s| = |V_t|$. |
| $K, I_{\text{in}}$ | Sinkhorn outer and inner iterations per outer iteration. |
| $\|\cdot\|_2$ | Euclidean norm. |
| $\|\cdot\|_F$ | Frobenius norm. |

