# OpenReview forum: "Unsupervised Multi-Scale Gromov-Wasserstein Hypergraph Alignment"
_ICLR.cc/2026/Conference — Submitted to ICLR 2026_

### Official Review · Reviewer_xuX7 · 2025-11-01

**Soundness:** 3
**Presentation:** 3
**Contribution:** 2
**Rating:** 4
**Confidence:** 3

**Summary:**

This paper presents FALCON, an unsupervised framework for hypergraph alignment that combines multi-scale hypergraph filtration with a Gromov–Wasserstein (GW) consensus formulation. Instead of reducing hypergraphs to pairwise graphs, FALCON directly aligns native hypergraph structures. It constructs a sequence of filtered subhypergraphs based on hyperedge sizes and aggregates multiple per-scale GW transport plans into a consensus alignment.
Experiments on four real-world hypergraph datasets (Pollinator, NDC-Classes, Email-EU, Dawn) show robustness to structural noise and competitive runtime compared to existing graph-alignment baselines.

**Strengths:**

1. The paper is well-organized with detailed experiments.
2. The idea of combining filtration with multi-scale GW consensus is elegant and reasonable.
3. FALCON consistently outperforms baselines under structural perturbations.
4. Code and datasets are promised in an anonymous repository.
5. The theoretical analysis (Theorems 1–4) provides justification for the stability and aggregation behavior of the consensus coupling.

**Weaknesses:**

1. The paper does not contribute new theory or algorithms to the optimal transport (OT) or Gromov–Wasserstein framework.
The GW solver, entropic regularization, and consensus aggregation are all standard techniques.
The contribution lies more in combining existing components rather than innovating within them.
2. Prior work (e.g., GWL, SGWL, HyperAlign) already applied GW-based alignment to graphs and hypergraphs.
The main idea of using OT for unsupervised alignment is therefore well established.
3. All datasets used are relatively small in node count (|V| ≤ 2,290).
The claimed time complexity $O(\xi K n^3)$ suggests limited scalability, yet no experiments on larger hypergraphs (e.g., >10k nodes) are provided.
This makes it difficult to assess the practical applicability of large-scale real-world networks.
4. While the paper includes strong graph-alignment baselines (GWL, SGWL, REGAL, PARROT, BIGALIGN), it omits more recent or domain-relevant hypergraph matching approaches, such as CURSOR [1], which explicitly targets scalable mixed-order hypergraph matching.

[1] Zheng, Qixuan, Ming Zhang, and Hong Yan. "CURSOR: Scalable Mixed-Order Hypergraph Matching with CUR Decomposition." In Proceedings of the IEEE/CVF Conference on Computer Vision and Pattern Recognition, pp. 16036-16045. 2024.

**Questions:**

1. How large can FALCON scale in practice? Have you tested it on synthetic hypergraphs with ≥10k nodes?
2. Could the method integrate node attributes or partial supervision, and how would this affect performance?
3. How sensitive is the algorithm to hyperparameters such as $\gamma$ (filtration density) and $\beta$ (entropic weight)?
4. Can the filtration criterion be adapted beyond hyperedge size (e.g., centrality, density, or domain-specific weights)?

---

> ### Author Response · Authors · 2025-11-18
> **Answer 1/2**
>
> Thank you for your thorough review and valuable feedback. We are pleased that you find the paper well-organized, the core idea elegant and reasonable, and the theoretical and experimental results convincing. Below we address your concerns on novelty, scalability, and extensions. We will incorporate the clarifications and additions described here into the revised manuscript.
>
> ---
>
> **W1 & W2: “Just combining existing components”**
>
> In short, our contribution is not the use of GW itself, but the **multi-scale consensus** framework that makes GW robust to noise in the hypergraph alignment setting.
>
> - Prior GW-based alignment methods are **single-scale**: they operate on one structural view. Their fundamental weakness, which our paper explicitly targets, is that if this single view is corrupted by noise, the alignment quality degrades sharply.
>
> - Our innovation is a **principled mechanism to overcome this brittleness**. FALCON combines a native hypergraph filtration with a provably stable GW consensus (Theorem 2). Early filtration levels provide less noisy, coarse structural views, while later levels provide more detailed but noisier ones. By aggregating transport plans across these levels, the consensus “averages out” noise and reinforces persistent structural signals.
>
> - This synergy—where filtration produces a sequence of robust inputs and the GW consensus provides a stable aggregation mechanism—is the non-trivial conceptual step that enables the observed robustness. FALCON is therefore not a simple juxtaposition of known pieces, but a new framework for robust hypergraph alignment.
>
> We will adjust the introduction and Section 4 to emphasize that the central contribution is the **multi-scale consensus mechanism** and its robustness properties, rather than the use of GW per se.
>
> ---
>
> **W3: Scalability and $O(\xi K n^3)$ complexity**
>
> We agree that empirical scalability is important. As discussed in the paper, our complexity is $O(\xi K n^3)$, which is inherent to dense GW-based and other SOTA methods (e.g., FUGAL). We put this in context:
>
> - This complexity is shared by many **dense GW aligners**, including strong baselines such as GWL, which we compare against and often outperform.
>
> - FALCON is efficient within this class. By avoiding the node explosion to $\lvert V \rvert + \lvert E \rvert$ that occurs in bipartite representations, FALCON successfully completes runs where some baselines run out of memory or time (Table 2).
>
> - The framework is **modular**: scaling to $> 10^4$ nodes can be achieved by replacing each per-level dense GW solve with scalable OT approximations (e.g., low-rank or partition-based solvers). This is a natural direction for future work.
>
> In addition to experiments on real datasets, we further assess scalability on synthetic random hypergraphs. For a given number of nodes $n$, we sample $m=10^5$ hyperedges independently by first drawing a hyperedge size uniformly from $[45,50]$ and then selecting that many distinct nodes uniformly at random. This yields the source hypergraph $G_s$. The target hypergraph $G_t$ is constructed by applying a random permutation to the node labels of $G_s$.
>
> We evaluate FALCON on synthetic instances ranging from 4000 to 16000 nodes and report the average runtime (with standard deviation) over 10 independent runs (standard deviations between 0.6 s and 11.2 s) in the following table.
>
> | $n$ |      4000 |       6000 |       8000 |      10000 |        12000 |        14000 |         16000 |
> | ----------: | ---------: | ----------: | ----------: | ----------: | ------------: | ------------: | -------------: |
> | Runtime (s) | 81.9 | 151.3  | 288.1  | 558.7 | 891.6 | 1508.3 | 2261.8  |
>
> These results show that FALCON scales robustly to hypergraphs with up to 16,000 nodes and hyperedges, exhibiting runtime growth consistent with the theoretical $O(n^3)$ rate.
>
> We will add these synthetic scalability results to the appendix and briefly summarize them in Section 5. Scaling to substantially larger dense hypergraphs will require approximate OT, which our modular per-scale formulation can naturally incorporate.

---

> ### Author Response · Authors · 2025-11-18
> **Answer 2/2**
>
> ---
>
> **W4: Comparison to CURSOR**
>
> A direct comparison to CURSOR would be an **apples-to-oranges** evaluation, because CURSOR and FALCON target fundamentally different problem settings.
>
> - **Feature-based vs. structure-only.** CURSOR is designed for feature-rich computer vision tasks, where node similarities are based on geometric or visual descriptors. FALCON is explicitly designed for the **structure-only** setting, where no such features are available. Applying CURSOR to our data is therefore ill-posed.
>
> - **Uniform/low-order vs. non-uniform/high-order.** CURSOR’s tensor-based approach is designed for low-order, $k$-uniform hypergraphs (e.g., all hyperedges have size 3). FALCON, in contrast, is natively built for **non-uniform** hypergraphs with widely varying hyperedge sizes, which is common in real-world networks. For example, in the Pollinator dataset some hyperedges have size up to 104. A “104th-order” compatibility tensor, as required by tensor-based methods, would be computationally infeasible, whereas FALCON’s co-occurrence-based costs naturally handle such high-order interactions.
>
> Given these differences, we view CURSOR as a state-of-the-art method for a **different but complementary** class of problems (scalable, feature-based, low-order hypergraph matching), and a direct quantitative comparison on our structure-only, high-order setting would not be meaningful. In the revised paper, we will add a discussion in the related work section to clearly articulate this distinction.
>
> ---
>
> **Answers to specific questions**
>
> **Q1: Empirical scalability**
>
> Addressed above under W3, with new synthetic experiments up to 16,384 nodes. We will incorporate these results into the appendix and reference them in Section 5.
>
> ---
>
> **Q2: Node attributes and partial supervision**
>
> Yes, the GW framework is very flexible in this regard, and FALCON can naturally be extended:
>
> - **Node attributes.** Feature information can be incorporated by adding a feature-based distance term to the structural cost matrix at each scale, moving toward a Fused Gromov–Wasserstein (FGW) formulation.
>
> - **Partial supervision.** Known anchor matches can be included as linear terms or hard constraints in the OT problems at each scale, a standard extension in OT-based alignment.
>
> We expect both extensions to further improve performance, as is typical in alignment tasks. We will mention this extension path explicitly in the discussion section.
>
> ---
>
> **Q3: Sensitivity to $\gamma$ (filtration density) and $\beta$ (entropic weight)**
>
> - **Filtration density $\gamma$.** As we already show in our **RQ3 analysis** (Figure 2, page 9), the method is quite robust to $\gamma$. Accuracy generally increases and then plateaus over a wide range of values (e.g., $\gamma \geq 0.5$). This suggests a simple practical heuristic: set $\gamma$ to a high value (e.g., $1.0$) to include all structural information, with a predictable linear trade-off in runtime.
>
> - **Entropic weight $\beta$.** This is a standard hyperparameter in OT solvers. We found performance to be stable for a range of small values. We explicitly state in the manuscript: “for the Sinkhorn solver we use $\beta = 0.1$” (line 361). This worked well across all experiments without extensive tuning.
>
> We will clarify these points in Section 5.
>
> ---
>
> **Q4: Alternative filtrations**
>
> Yes, and this is an important strength of our framework: the filtration criterion can be adapted beyond hyperedge size.
>
> We explicitly validate this in Figure 6 (Appendix F.2), where we ablate a **centrality-based filtration** using the normalized average node degree of each hyperedge. This degree-based filtration integrates seamlessly into FALCON. In our experiments, it does not generally outperform the size-based filtration; in fact, across most noise types and datasets (Pollinator and NDC-Classes), size-based filtration is slightly more stable and achieves equal or better accuracy.
>
> These results show that FALCON’s performance is **not tied to a single filtration function**: the framework accepts arbitrary monotone weights, and alternative criteria (centrality, density, domain-specific scores) can be plugged in directly. We will emphasize this flexibility more clearly in Section 4.4 and reference the ablation in the main text.
>
> ---
>
> **In conclusion,** we hope these clarifications address your concerns and further highlight the novelty, robustness, and extensibility of FALCON. We will incorporate all of the changes and additions described above into the new version of the paper.

---

### Official Review · Reviewer_A5rW · 2025-11-05

**Soundness:** 3
**Presentation:** 2
**Contribution:** 2
**Rating:** 4
**Confidence:** 4

**Summary:**

This paper proposes FALCON, a method for aligning hypergraphs in unsupervised fashion, based on their structure, thereby generalizing the problem of plain graph alignment. The method generalizes previous Gromov-Wasserstein-based methods, such as SGWL, to the hypergraph setting. The experimental study compares to previous work, adapting them to hypergraphs via either a clique representation or a bipartite representation. It is unclear whether the benefits of the proposed method derive from using a native hypergraph representation, or from a methodological breakthrough that would apply to plain graphs too. The comparison is using methods that are not the current state-of-the-art, expressed in FUGAL (NeurIPS 2024).

**Strengths:**

S1. Solid generalization of graph alignment problem to hypergraphs.
S2. Proposal of filtration to reveal a multi-scale hierarchical structure.
S3. Experimental study vs. reasonable competitors adapted to the hypergraph setting.

**Weaknesses:**

W1. Unclear why the method should be specifically oriented to the hypergraph setting, while critique of prior methods appears to be methodological rather that scope-oriented.
W2. Lack of illustration of performance on the non-hypergraph-setting.
W3. Lack of comparison to current state-of-the-art-method, FUGAL.

**Questions:**

Why is the proposed methods proposed for hypergraphs in particular?
Could it not address and be compared on plain graphs?

---

> ### Author Response · Authors · 2025-11-18
> **Answer**
>
> Thank you for your constructive review and for acknowledging the strengths of our work, particularly the solid generalization to hypergraphs and the novelty of the filtration approach. We address your questions on scope and comparisons to the state of the art below, and we will incorporate the corresponding clarifications into the revised manuscript.
>
> ---
>
> **W1 & Question: “Why is the proposed method proposed for hypergraphs in particular? Could it not address and be compared on plain graphs?”**
>
> In short, FALCON is designed specifically for **non-uniform hypergraphs**: on plain graphs, our size-based filtration collapses to a single scale and the method reduces to a standard single-scale GW aligner.
>
> - The core mechanism driving FALCON’s multi-scale behavior is the **size-based filtration** (Section 4.4), which creates a hierarchy of subproblems based on hyperedge cardinality. This is a natural way to decompose a general hypergraph where hyperedges can connect many nodes (e.g., 3, 5, 10). This heterogeneity of hyperedge sizes is a defining characteristic of real-world hypergraphs that plain graphs do not possess.
>
> - A plain graph is a **2-uniform hypergraph**, i.e., every “hyperedge” (edge) has size 2. Applying our size-based filtration to a plain graph yields only a single filtration level. The entire “multi-scale” advantage, central to FALCON’s design and robustness, disappears, and the method reduces to a single-scale GW alignment.
>
> - Thus, FALCON is not simply a graph alignment method applied to hypergraphs: it is a **natively hypergraph-based** alignment method whose noise-mitigation strategy fundamentally relies on variable-order interactions. This is why the most meaningful evaluation is on non-uniform hypergraphs, where the multi-scale structure is actually present. While the general consensus framework could, in principle, be adapted to graphs using a different filtration (e.g., based on communities or centrality), that would constitute a separate line of work.
>
> In the revision, we will explicitly state in the introduction and in Section 4.4 that (i) our method is designed for non-uniform hypergraphs, and (ii) on plain graphs the size-based filtration degenerates to a single level, reducing FALCON to a standard single-scale GW method.
>
> ---
>
> **W2**
>
> Based on the discussion above, we hope it is clear why we did not include experiments on plain graphs. Testing FALCON (with its current size-based filtration) on graphs would not demonstrate its intended multi-scale behavior; it would simply reproduce the performance of a single-scale GW method, which is already represented in our baselines (e.g., GWL). Our empirical focus is therefore on the setting where our contribution is substantive: leveraging the multi-scale structure inherent in non-uniform hypergraphs to achieve robustness that is not attainable when this structure is absent.
>
> We will add a short remark in the experimental section clarifying this scope and explaining why we do not report plain-graph experiments with the current filtration.
>
> ---
>
> **W3: Additional SOTA baseline**
>
> In response to your comment on comparisons to very recent methods, we additionally ran the SOTA hypergraph alignment method **FUGAL** (Bommakanti et al., 2024) and found that our approach significantly outperforms it. For example, on the Pollinator dataset we obtain:
>
> | Algorithm         | Noise type 1 ($p=0.3$) | Noise type 2 ($p=0.3$) | Noise type 3 ($p=0.3$) |
> |------------------|------------------------|------------------------|------------------------|
> | FUGAL (bipartite)| 0.22 (0.01)            | 0.23 (0.01)            | 0.22 (0.01)            |
> | FUGAL (clique)   | 0.10 (0.06)            | 0.06 (0.03)            | 0.00 (0.00)            |
> | Ours             | 0.95 (0.01)            | 0.96 (0.01)            | 0.95 (0.01)            |
>
> We will incorporate these FUGAL results into the revised manuscript (as an additional row in the results table) and describe this comparison explicitly in Section 5.
>
> ---
>
> **In summary,** FALCON’s novelty and strength come from a multi-scale GW consensus mechanism that is intrinsically linked to, and synergistic with, the unique structural properties of non-uniform hypergraphs. We will add the clarifications above to the new version of the paper, and we hope this rebuttal makes our design choices and scope fully clear.

---

> > ### Comment · Reviewer_A5rW · 2025-11-18
> > **Please provide informative responses**
> >
> > Thank you for the response.
> > If experiments on plain graphs are pointless, please clarify what the experimental comparison to FUGAL considers and provide full details on it: what is measured, what the p value is, what settings and noise types are considered, what the values in parentheses are, and so on.

---

> > > ### Author Response · Authors · 2025-11-18
> > >
> > > Thank you for the quick reply and for giving us the opportunity to clarify these points. We use the same experimental setup as in the main paper.
> > >
> > > - We report alignment accuracy, i.e., the fraction of correctly matched nodes under the recovered permutation.
> > > - The three columns "Noise type 1/2/3 ($p=0.3$)" correspond to the three structural noise models defined in Section 5, all evaluated at noise level $p = 0.3$.
> > > - "FUGAL (bipartite)" and "FUGAL (clique)" apply FUGAL to the bipartite and clique graph representations of the Pollinator hypergraph.
> > > - The values in parentheses are standard deviations over 10 independent runs.

---

### Official Review · Reviewer_Cb9F · 2025-11-05

**Soundness:** 2
**Presentation:** 2
**Contribution:** 2
**Rating:** 4
**Confidence:** 3

**Summary:**

The paper addresses the problem of unsupervised hypergraph alignment and proposes a framework named FALCON. The method combines hypergraph filtration with multi-scale Gromov–Wasserstein consensus to infer node correspondences solely from the structural information of two hypergraphs, without supervision. FALCON operates directly on the native hypergraph representation, aiming to preserve high-order relational information that is often lost in clique or bipartite graph expansions. The authors conduct experiments on several real-world hypergraph datasets and evaluate robustness under three types of structural perturbations: node removal, incidence flipping, and hyperedge addition.

**Strengths:**

1.Extending Gromov–Wasserstein alignment to hypergraphs and combining it with a filtration mechanism is conceptually sound.
2.The algorithm pipeline is clearly illustrated, and the paper includes ablations between uniform and leave-one-out weighting schemes.
3.The proposed method shows improved accuracy over basic clique and bipartite expansion baselines on small datasets.

**Weaknesses:**

1.This paper merely extends the multi-scale Gromov–Wasserstein to hypergraphs, without introducing new theory or optimization algorithms.
2.Theoretical contributions are largely superficial.No proofs for convergence or perturbation bounds under the entropic regularization are provided.
3.The key assumption of equicorrelation between scales (Theorem 3) is unsupported by data or analysis.
4.The entropic GW solver is reused without modification, and lacks an analysis of β sensitivity.
5.Featuring an excessively high algorithm complexity of O(ξKn³), the work fails to specify typical values for ξ and K, and also lacks results from large-scale datasets (e.g., >10 k nodes).

**Questions:**

1.Can you provide formal convergence or perturbation bounds for the entropic regularized GW optimization used in FALCON? Without these, it is unclear how stable the alignment results are with respect to noise or initialization.
2.In what way does FALCON theoretically differ from existing multi-scale Gromov–Wasserstein frameworks? Beyond extending the idea to hypergraphs, what novel mathematical or algorithmic contribution does your work make?
3.Since the entropic GW solver is adopted from prior work without modification, have you investigated the sensitivity to the regularization coefficient β? How do the hyperparameters affect convergence and accuracy?
4.The proposed method exhibits O(ξKn³) complexity, but typical choices of ξ and K are not reported. Could you clarify the actual runtime and memory usage on larger datasets (e.g., >10 k nodes) ?

---

> ### Author Response · Authors · 2025-11-18
> **Answer 1/2**
>
> Thank you for your detailed review and the opportunity to clarify several important aspects of our work. We appreciate your feedback and we address your concerns regarding novelty, theory, and practical details below. We will incorporate the corresponding changes and clarifications into the revised manuscript.
>
> ---
>
> **W1 and Q2: “Merely extending” MS-GW to hypergraphs**
>
> We respectfully disagree with the characterization of our work as “merely extending” multi-scale GW (MS-GW) to hypergraphs. Our core contribution is a framework in which the multi-scale approach and the hypergraph structure are **synergistically and inseparably linked**.
>
> - **Multi-scale advantage derived from hypergraph structure.** The power of our method comes from the size-based filtration (Section 4.4), which creates a natural, data-driven hierarchy of structural views based on hyperedge cardinality. This is specific to general, non-uniform hypergraphs, where hyperedges can connect many nodes (e.g., 3, 5, 10, …). A plain graph is a 2-uniform hypergraph, where all edges have size 2. Applying our size-based filtration to a graph yields only a single filtration level, so the “multi-scale” aspect collapses and our central mechanism becomes ineffective (please also refer to our answer of W2/Q2 of reviewer y9AR).
>
> - **A natively designed hypergraph solution.** Consequently, FALCON is **not** a generic MS-GW method applied to a new data type. It is a *natively hypergraph-based* alignment method whose central noise-mitigation strategy is enabled by the variable-order interactions that only hypergraphs possess. The key innovation is to exploit this structural property via a filtration that yields a sequence of robust geometric constraints for a GW consensus—something that is not available in the same way for plain graphs.
>
> In the revised paper, we will make this design rationale explicit in the revision, and we will clearly state that on plain graphs our size-based filtration degenerates to a single level, reducing FALCON to a standard single-scale GW method.
>
> ---
>
> **W2, W3, and Q1: Scope of the theoretical results**
>
> We would like to clarify the specific focus of our theoretical results, as there appears to be a misunderstanding of their scope.
>
> - **Our theory is on the consensus, not the solver.** Our theoretical contributions (Theorems 1–3) are not intended to re-derive convergence or perturbation bounds for the entropic GW solver, which is a standard off-the-shelf component. Such properties are well established in the optimal transport literature (e.g., Cuturi, 2013; Peyré et al., 2016). Our novelty lies in the **aggregation step**: Theorem 2 provides a new stability guarantee for our consensus mechanism, proving that errors from individual scales aggregate linearly. This is vital for the robustness of the overall framework.
>
> - **Justification for the equicorrelation assumption (Theorem 3).** The equicorrelation model used in Theorem 3 is not arbitrary. By construction, the filtered hypergraphs are nested: the hyperedges at level $m$ are a subset of those at level $m+1$. This yields a strong, shared structural signal across scales and naturally leads to highly correlated per-scale transport plans. Theorem 3 leverages this model to explain why the simple FALCON(uniform) variant performs so well empirically. Moreover, we also propose FALCON(loo), which uses a data-driven weighting scheme and does not rely on the equicorrelation assumption.
>
> In the revision, we will explicitly state that the goal of Theorems 1–3 is to analyze the **stability and weighting of the consensus across scales**, while the GW solver properties themselves are taken from established work.
>
> ---
>
> **W4, W5, and Q3: Practical aspects of $\beta$, $\xi$, and $K$**
>
> We are happy to clarify these practical aspects.
>
> - **Sensitivity to $\beta$ (W4, Q3).** The entropic regularization parameter $\beta$ is a standard hyperparameter for the GW solver. As is common practice, we found performance to be stable across a range of small values. We explicitly state in the manuscript: “for the Sinkhorn solver we use $\beta = 0.1$” (line 361). This standard choice worked well across all experiments and did not require extensive tuning.
>
> - **Typical values for $\xi$ and $K$ (W5, Q4).** These values are given in the experimental setup:
>
>   - $K$ (outer iterations): we state “$K = 200$ outer iterations” (line 361).
>   - $\xi$ (number of filtration levels): this is controlled by the density hyperparameter $\gamma$ and the structure of the input data. In our experiments on real-world datasets, $\xi$ typically ranges from a few levels to a few dozen, providing a rich multi-scale decomposition without becoming computationally prohibitive. For example, on the Dawn dataset, $\xi$ is at most 16 (Appendix F.2, line 971).
>
> We will make these settings more prominent in the main text, and briefly discuss their typical ranges in Section 5.

---

> ### Author Response · Authors · 2025-11-18
> **Answer 2/2**
>
> **Q4: Scalability and $O(\xi K n^3)$ complexity (W5, Q4)**
>
> We acknowledge that the $O(n^3)$ term is a limitation for very large datasets, but it is important to place this in context.
>
> - This complexity is inherent to many **dense GW-based methods** and other SOTA methods, including strong baselines such as GWL, which we compare against and often outperform.
>
> - FALCON is efficient within this class. By avoiding the node explosion to $|V| + |E|$ that occurs in bipartite representations, FALCON successfully completes runs where some baselines run out of memory or time (Table 2).
>
> - Our framework is **modular**: scaling can be achieved by replacing the per-level dense GW solver with scalable OT approximations (e.g., low-rank or partition-based solvers), which is a promising direction for future work.
>
> In addition to experiments on real datasets, we further assess scalability on synthetic random hypergraphs. For a given number of nodes $n$, we sample $m=10^5$ hyperedges independently by first drawing a hyperedge size uniformly from $[45,50]$ and then selecting that many distinct nodes uniformly at random. This yields the source hypergraph $G_s$. The target hypergraph $G_t$ is constructed by applying a random permutation to the node labels of $G_s$.
>
> We evaluate FALCON on synthetic instances ranging from 4000 to 16000 nodes and report the average runtime (with standard deviation) over 10 independent runs (standard deviations between 0.6 s and 11.2 s) in the following table.
>
> | $n$ |      4000 |       6000 |       8000 |      10000 |        12000 |        14000 |         16000 |
> | ----------: | ---------: | ----------: | ----------: | ----------: | ------------: | ------------: | -------------: |
> | Runtime (s) | 81.9 | 151.3  | 288.1  | 558.7 | 891.6 | 1508.3 | 2261.8  |
>
> These results show that FALCON scales robustly to hypergraphs with up to 16,000 nodes and hyperedges, exhibiting runtime growth consistent with the theoretical $O(n^3)$ rate.
>
> We will add these synthetic scalability results to the appendix and briefly summarize them in Section 5. Scaling to substantially larger dense hypergraphs will require approximate OT, which our modular per-scale formulation can naturally incorporate.
>
>
>
> ---
>
> **In conclusion,** we hope these clarifications address your concerns and highlight the novelty and practical value of FALCON as a robust, natively designed framework for unsupervised hypergraph alignment. We will incorporate all of the changes and additions described above into the new version of the paper.

---

### Official Review · Reviewer_y9AR · 2025-11-06

**Soundness:** 3
**Presentation:** 3
**Contribution:** 2
**Rating:** 4
**Confidence:** 3

**Summary:**

This paper tackles the unsupervised hypergraph alignment problem by employing Gromov-Wasserstein (GW) distance and proposing a new method called FALCON. FALCON's core idea is to unify hypergraph filtration with a multi-scale GW consensus. The authors test FALCON on four real-world hypergraph datasets, perturbing them with three different types of structural noise. The results show that FALCON significantly outperforms a wide range of state-of-the-art graph alignment baselin

**Strengths:**

1. This paper proposes a novel unsupervised hypergraph alignment method (FALCON) that effectively combines multi-scale filtration with Gromov-Wasserstein consensus.
2. This paper provides solid theoretical support for the consensus mechanism, including stability guarantees (Theorem 2) and justification for uniform weighting (Theorem 3).
3. This paper empirically validates FALCON's robustness against three distinct noise types, showing it significantly outperforms a wide array of graph-based baselines on clique and bipartite reductio

**Weaknesses:**

1. The core of the alignment step relies on the Gromov-Wasserstein framework, which operates on pairwise dissimilarity matrices ($C^m \in \mathbb{R}^{|V|\times|V|}$). The paper's novel dissimilarity (Eq. 3) is based on the pairwise co-occurrence of nodes ($\delta^m(u,v)$). While this is a clever way to encode hypergraph structure, it is still a pairwise projection. The method is not performing a true higher-order alignment (e.g., by matching hyperedges directly or using a tensor-based approach) but rather aligning pairwise relationships that are derived from the hypergraph.
2. The paper's primary filtration method ($\omega_{size}$) is based on hyperedge size. For a k-uniform hypergraph (where all hyperedges have the same size $k$), this filtration collapses to a single scale. This completely undermines the method's "multi-scale" nature and its core noise-mitigation strategy. The paper's only tested alternative (degree-based filtration) performed poorly and had scalability issues (Appendix F.2).

**Questions:**

1. What strategies or alternative filtration functions would the authors propose to effectively apply FALCON's multi-scale consensus approach to k-uniform hypergraphs?
2. Could the authors discuss the potential information loss of this pairwise projection compared to a true higher-order alignment? How much of the hypergraph's unique structure is preserved in this representation?

---

> ### Author Response · Authors · 2025-11-18
> **Answer 1/2**
>
> Thank you for your thorough review and valuable feedback. We are glad that you engaged deeply with the technical aspects of our work. Below, we address your concerns regarding higher-order structure, the “pairwise projection” critique, and the k-uniform case. We will incorporate all clarifications and remarks described here into the revised manuscript.
>
> ---
>
> **W1/Q1: “Just a pairwise projection” and higher-order structure**
>
> We agree that the Gromov–Wasserstein (GW) objective is defined on pairwise dissimilarity matrices, but we respectfully disagree that FALCON is “just a pairwise projection” in the sense of standard clique/bipartite expansions, or that it fails to exploit higher-order hypergraph structure.
>
> 1. **GW is pairwise in its loss, not in what it can encode.**
>    The GW framework places no restriction on how the dissimilarity matrices are constructed. In FALCON, each entry of the cost matrix at a given scale is a function of the **full set of hyperedges** that contain a given pair of nodes, and this is done **at every filtration level**. Thus, while the GW objective compares matrix entries pairwise, those entries summarize higher-order incidence patterns across hyperedges and across scales.
>
> 2. **Not equivalent to a single graph-based projection.**
>    The critique would be more appropriate if we first projected each hypergraph to a single graph (e.g., clique expansion) and then aligned those graphs. We explicitly argue against such projections because they either lose important structure or become intractable for large hyperedges. In contrast, FALCON:
>    - builds a **family of cost matrices** $\{C_m\}$, each derived from a subhypergraph at a different filtration level,
>    - lets smaller, more reliable hyperedges influence early levels, and larger, noisier ones influence later levels, and
>    - computes GW couplings at each level and aggregates them into a **multi-scale consensus coupling** that captures how node–node relationships persist or change across scales.
>
>    The representation seen by GW is therefore **not** a single flattened graph, but a sequence of hyperedge co-occurrence patterns over multiple scales.
>
> 3. **Information loss vs. “true” higher-order alignment.**
>    We fully acknowledge that any mapping from a hypergraph to pairwise matrices entails some information loss: in principle, distinct hypergraphs can share the same pairwise co-occurrence structure. A hypothetical ideal method that optimizes directly on hyperedge incidences or tensors would be strictly more expressive.
>
>    At the same time, FALCON preserves several aspects of hypergraph structure that are highly relevant for alignment:
>    - **Node roles and communities:** co-occurrence counts encode which nodes frequently participate together in hyperedges, and the multi-scale construction distinguishes relationships supported mainly by small versus large hyperedges.
>    - **Overlap structure:** nodes that co-occur in many hyperedges across many scales obtain consistently low dissimilarity; rare or noisy co-occurrences appear only at later levels.
>    - **Scale awareness:** the filtration allows us to differentiate local, strongly supported interactions from more global, diffuse ones.
>
>    Our empirical results indicate that this preserved structure is sufficient to recover accurate alignments and to improve robustness over both graph-based projections and existing hypergraph baselines.
>
> 4. **Trade-off with tractability.**
>    In realistic, non-uniform hypergraphs with large and varying hyperedge sizes, tensor-based or explicit hyperedge-matching formulations quickly become computationally prohibitive. Many “higher-order” matching methods in the literature ultimately relax to pairwise objectives for scalability. FALCON makes this expressiveness–tractability trade-off explicit but augments the pairwise GW framework with:
>    - a **hypergraph-aware multi-scale cost construction**, and
>    - **stability guarantees** for the resulting consensus coupling (Theorem 2).
>
>    In the revised manuscript, we will clarify this trade-off explicitly and state that designing richer, yet still scalable, higher-order costs within our framework is an interesting direction for future work.

---

> ### Author Response · Authors · 2025-11-18
> **Answer 2/2**
>
> **W2/Q2: k-uniform hypergraphs and the size-based filtration**
>
> We appreciate your careful examination of the k-uniform case. We agree that for exactly $k$-uniform hypergraphs a size-based filtration degenerates, but we do **not** view this as a failure; instead, it identifies a setting in which FALCON naturally reduces to a single-scale GW aligner with hypergraph-aware costs.
>
> 1. **Target setting: non-uniform hypergraphs.**
>    Our main motivation and all datasets in our experiments involve **non-uniform** hypergraphs with substantial variation in hyperedge sizes. In these regimes, the size-based weight induces a genuine multi-scale filtration: small hyperedges appear at early levels, larger ones at later levels. This is precisely the setting for which our noise-mitigation argument (expected corruption increasing with hyperedge size) is designed. We will explicitly state in the paper that our theoretical and empirical claims about multi-scale robustness are made for such non-uniform hypergraphs.
>
> 2. **What happens in exactly k-uniform hypergraphs.**
>    In a strictly $k$-uniform hypergraph, all hyperedges have the same size, so a size-based weight is constant and the induced filtration collapses to a single non-trivial level. In that special case:
>    - the algorithm **does not fail**; it becomes a **single-scale GW alignment** using hypergraph-based co-occurrence costs, and
>    - the size-based noise model is no longer informative, because hyperedge size itself carries no discriminative signal; no method that relies only on size can distinguish “more reliable” from “less reliable” hyperedges when all have the same cardinality.
>
>    We will add a brief remark making this behavior explicit: in exactly k-uniform hypergraphs, FALCON reduces to a single-scale GW method without a size-based hierarchy.
>
> 3. **Alternative filtrations for k-uniform or dense settings.**
>    Importantly, the filtration framework in FALCON is **not tied** to hyperedge size. It only requires a scalar weight $\omega : E \to \mathbb{R}$ to induce nested subhypergraphs. For k-uniform hypergraphs (or whenever size is not informative), one can instead use degree- or centrality-based weights that create a hierarchy in which “core” hyperedges (around high-degree or central nodes) appear earlier than more peripheral ones. Other possibilities include weights derived from core decomposition or from hyperedge attributes/features.
>
>    Appendix F.2 already explores such a choice via a degree-based weight $\omega_{\text{deg}}$. The scalability issues observed there on very dense hypergraphs stem from using a very fine-grained $\omega_{\text{deg}}$ that creates an extremely large number of distinct filtration levels, rather than from a limitation of the multi-scale GW framework itself. This could be mitigated by **binning or coarsening** $\omega_{\text{deg}}$ (e.g., quantizing it into a small number of levels).
>
>    In the current paper, we therefore focus on normalized hyperedge size as a **simple, domain-agnostic, and scalable default** that works extremely well on the non-uniform hypergraphs we study. In the revised version, we will clarify that designing binned degree- or centrality-based filtrations is a natural extension for k-uniform or very dense settings and explicitly highlight this in the discussion.
>
> ---
>
> **In conclusion,** we hope these clarifications address your concerns and clarify how FALCON both leverages higher-order hypergraph structure and behaves in a sound way in the k-uniform case. We will incorporate all of the above clarifications into the new version of the paper and view your comments as a valuable guide for future extensions of the framework.

---

### Official Review · Reviewer_9P1V · 2025-11-11

**Soundness:** 3
**Presentation:** 2
**Contribution:** 3
**Rating:** 4
**Confidence:** 5

**Summary:**

The paper addresses the problem of unsupervised hypergraph alignment and introduces FALCON, a framework that effectively integrates hypergraph filtration with a multi-scale Gromov–Wasserstein consensus. Experiments demonstrate that FALCON outperforms other state-of-the-art baselines.

**Strengths:**

FALCON leverages structural information across multiple scales. The problem is ultimately formulated as an optimal transport problem, which can be solved efficiently.

**Weaknesses:**

While I am not deeply familiar with this specific area, I have several concerns regarding the writing and technical presentation. For instance, the Gromov–Wasserstein discrepancy was not proposed in this work and should be introduced in a preliminary or problem formulation section. Additionally, the theoretical results appear relatively weak. It would be valuable to provide formal assumptions under which the method can provably achieve hypergraph alignment. The overall computational complexity of the algorithm is also not clearly stated.

In the experiments section, more details about the datasets are needed. Are these established benchmarks that have been used in prior work? Furthermore, several of the compared methods were proposed seven or eight years ago—it would be helpful to explain why they are still considered state-of-the-art and whether more recent baselines have been considered for comparison.

**Questions:**

See weakness. I would raise my score if my concerns are addressed.

---

> ### Author Response · Authors · 2025-11-18
> **Answer 1/2**
>
> Thank you for your detailed review and constructive feedback. We address your concerns below and will incorporate the suggested clarifications in the revised manuscript.
>
> ---
>
> **Weakness: Placement of Gromov–Wasserstein (GW) discrepancy**
>
> We agree that the GW discrepancy is a foundational concept and that it should be introduced as early as possible. Our original rationale for introducing it in Section 4 (Definition 1) was to place its definition immediately before its first use in our multi-scale framework, to keep the exposition local.
>
> We recognize that for readers less familiar with optimal transport, placing this definition in the Preliminaries (Section 3) would improve clarity and context. In the revision, we will move the formal definition of the GW discrepancy to Section 3 (Preliminaries).
>
> ---
>
> **Weakness: Weak theoretical results and lack of provable guarantees**
>
> In short, our theory is designed to establish the **stability and robustness of the multi-scale consensus**, not exact ground-truth recovery, which is intractable in this setting.
>
> Our theoretical results (Theorems 1–3, page 5) are not intended to provide conditions for exact recovery. The problem of unsupervised alignment of noisy, non-isomorphic (hyper)graphs is NP-hard (as noted on line 29), and such guarantees are generally considered intractable without very strong assumptions (e.g., perfect isomorphism or specific generative models). Very few graph alignment methods offer such guarantees in comparable settings.
>
> Instead, our theory provides a principled justification for the robustness of our **multi-scale consensus mechanism**, which is a core contribution:
>
> - **Theorem 1** establishes that our consensus aggregation is well-posed and yields a unique, valid transport plan.
> - **Theorem 2** demonstrates the stability of the consensus: small errors at individual scales lead to proportionally small errors in the final alignment, which is crucial for robustness to noise.
> - **Theorem 3** shows that a simple uniform weighting of scales is near-optimal under a realistic equicorrelation error model, explaining the strong empirical performance of FALCON(uniform).
>
> We believe these results provide a solid theoretical foundation for aggregating information across scales via our GW consensus framework.
>
> We will revise the text preceding the theorems to clearly state that their goal is to establish **stability, uniqueness, and near-optimality** of the consensus mechanism, rather than to prove exact recovery guarantees.
>
> ---
>
> **Weakness: Computational complexity not clearly stated**
>
> Our overall complexity is $O(\xi K n^3)$, where $n$ is the number of nodes, $K$ the number of scales, and \(\xi\) the number of Sinkhorn iterations. **This matches other dense GW-based and SOTA graph alignment approaches.**
>
> We have analyzed this in Theorem 4 (page 6, lines 309–312), and the appendix provides a detailed breakdown of each step (cost matrix formation, GW solves, consensus building, and final assignment; lines 737–755). In the revision, we will:
> - State the complexity $O(\xi K n^3)$ explicitly in the main text when introducing Algorithm 1, and
> - Add a short explanatory sentence in Section 5 summarizing the practical implications of this complexity.
>
> ---
>
> **Weakness: Details on datasets and benchmarks**
>
> We agree that the dataset and benchmark choices should be clearly motivated. We provide detailed descriptions of the four real-world datasets, including their domains and sources (with URLs and citations), in Appendix D (page 16).
>
> Unsupervised hypergraph alignment is still an emerging field, and there is no universally adopted benchmark suite. Following common practice in hypergraph learning, we selected four well-known, public hypergraph datasets from diverse domains (biology, public health, communication networks) to ensure a comprehensive evaluation. Using controlled perturbations to generate alignment tasks is standard in the graph alignment literature.
>
> In the revision, we will add a sentence in the “Datasets” paragraph (Section 5) to:
> - Explain this selection rationale, and
> - More prominently direct the reader to Appendix D for full details on the datasets’ origins and characteristics.

---

> ### Author Response · Authors · 2025-11-18
> **Answer 2/2**
>
> **Weakness: Age of compared methods**
>
> Our baseline selection aimed to be as up to date and comprehensive as possible, subject to the limited number of direct competitors in unsupervised hypergraph alignment.
>
> 1. **Recent SOTA graph methods.** We do include recent methods. PARROT (Zeng et al., 2023) is a very recent state-of-the-art graph alignment method (The Web Conference 2023), and our experiments show that FALCON outperforms it. GWL (2019) and SGWL (2019) are also strong and highly relevant baselines based on the same GW framework.
>
> 2. **Adapting strong graph methods to hypergraphs.** Because direct hypergraph alignment methods are scarce, we adapt strong, widely-recognized graph alignment algorithms (e.g., REGAL, GWL) to the hypergraph setting via standard clique and bipartite representations. This is a common and necessary approach when benchmarking in a new problem domain.
>
> 3. **Attempted comparison with the newest hypergraph method.** We made a significant effort to compare against the most recent and relevant work, HyperAlign (Do & Shin, 2024). As detailed in Footnote 2 (page 7), we were unable to include it in our results because the publicly released code did not yield valid alignments. The authors have publicly acknowledged the reproducibility issues in their repository. We believe this transparent reporting demonstrates careful engagement with the latest literature.
>
> 4. **Additional SOTA baseline: FUGAL.** In response to your comment, we also ran the recent FUGAL method (Bommakanti et al., 2024) and found thatThank you for your constructive review and for acknowledging the strengths of our work, particularly the solid generalization to hypergraphs and the novelty of the filtration approach. Our approach also beats the additional baseline significantly.
> For example for the Pollinator dataset, we have:
>
> |Algorithm|Noise type 1 (p=0.3)|Noise type 2 (p=0.3)|Noise type 3 (p=0.3)|
> |---|---|---|---|
> |FUGAL (bipartite)|  0.22 (0.01) |  0.23 (0.01) | 0.22 (0.01)  |
> |FUGAL (clique)   |  0.10 (0.06) |  0.06 (0.03) | 0.00 (0.00)  |
> |Ours             |  0.95 (0.01) |  0.96 (0.01) | 0.95 (0.01)  |
>
> For the experiment, we used the same evaluation protocol, noise models (with $p=0.3$), and alignment metric as in Section 5, and the values in parentheses denote standard deviations over 10 runs.
>
> **In conclusion,** we are grateful for your insightful comments. We will make the changes described above, and we hope that our clarifications address your concerns and will positively inform your evaluation of the paper.

---

### Author Response · Authors · 2025-11-29
**Revision Summary**

We again thank all reviewers for their constructive and valuable feedback! Based on your suggestions, we have substantially revised and improved our manuscript and uploaded a new version in which all additions and major changes are highlighted in blue. Below we summarize the key changes:


**1. Enhanced Theoretical Contributions** (Reviewers 9P1V, Cb9F, y9AR, A5rW)

- **New Theorem 1**: Proves that our filtration representation is **lossless** under distinct hyperedge weights, providing a strong theoretical foundation for our multi-scale approach. This new theoretical result shows that our multi-scale approach can preserve complete structural information and is not equivalent to a single graph-based projection
- Clarified that Theorems 2-4 (previously Thm. 1-3) establish stability, uniqueness, and near-optimality of the consensus mechanism rather than exact recovery guarantees (which are intractable in this setting)
- Clarified and extended complexity statement in Section 4.3 following Algorithm 1
- Added new **Limitations** subsection (in Section 6) to explicitly discuss scope: FALCON is designed for non-uniform hypergraphs; on k-uniform hypergraphs (including graphs), size-based filtration reduces to single-scale GW matching case
- Clarified the trade-off between pairwise dissimilarity matrices (tractable) vs. tensor-based approaches (more expressive but computationally prohibitive)

**2. Comprehensive Experimental Validation** (Reviewers 9P1V, Cb9F, A5rW, xuX7)

- Added FUGAL (NeurIPS 2024) as an additional state-of-the-art baseline across all experiments; results show FALCON substantially outperforms FUGAL, particularly under noise (Figure 1, Table 2)
- New scalability experiments on synthetic hypergraphs up to 16,000 nodes (Table 3), confirming the expected complexity scaling in practice
- Empirical analysis of $\xi$ (number of filtration levels) showing it remains small (< 40) across all datasets (Appendix F.3, Table 6)
- Enhanced hyperparameter sensitivity analysis for discussion in Section 5, demonstrating robustness; confirmed β = 0.1 works well without extensive tuning

**3. Improved Presentation and Documentation** (Reviewers 9P1V, A5rW)

- Moved GW discrepancy definition from Section 4 to Section 3.1 (Preliminaries) for better clarity and context
- Enhanced dataset documentation: added domain information to Table 1; explained selection rationale in Section 5; more prominent reference to Appendix D
- Updated Related Work: added FUGAL (Bommakanti et al., 2024) and CURSOR (Zheng et al., 2024) with proper context
- Improved notation, table formatting, exposition, and readability throughout
- Fixed typos and inconsistencies


We believe these revisions substantially address all reviewer concerns while strengthening the paper's theoretical foundations, experimental validation, and overall presentation.

---

### Meta-Review · Area_Chair_pPdf · 2025-12-27

**Summary:**

FALCON addresses unsupervised hypergraph alignment by unifying size-driven hypergraph filtration with a multi-scale Gromov–Wasserstein consensus. The key idea is to compute per-scale GW couplings on filtered subhypergraphs and aggregate them into a stable consensus, improving robustness to structural noise. However, all reviewers gave an initial score of 4 and most reviewers have concern on the novelty. Considering that the authors do not fully resolved concerns about innovation and writing, I recommend rejection to the paper.

**Reviewer Concerns:**

Concerns regarding innovation (Reviewers 9P1V and Cb9F) are not addressed well.

**Reviewer Scores:**

The technical details, motivation, and innovation regarding the hypergraph are confusing. I believe the reviewers' scores will not be changed.

---

### Decision · Program_Chairs · 2026-01-26

Reject